# Molecular phylogenies map to biogeography better than morphological ones

Jack W. Oyston [1 ✉], Mark Wilkinson[2], Marcello Ruta [3] & Matthew A. Wills[1 ✉]

Phylogenetic relationships are inferred principally from two classes of data: morphological and molecular. Currently, most phylogenies of extant taxa are inferred from molecules and when morphological and molecular trees conflict the latter are often preferred. Although supported by simulations, the superiority of molecular trees has rarely been assessed empirically. Here we test phylogenetic accuracy using two independent data sources: biogeographic distributions and fossil first occurrences. For 48 pairs of morphological and molecular trees we show that, on average, molecular trees provide a better fit to biogeographic data than their morphological counterparts and that biogeographic congruence increases over research time. We find no significant differences in stratigraphic congruence between morphological and molecular trees. These results have implications for understanding the distribution of homoplasy in morphological data sets, the utility of morphology as a test of molecular hypotheses and the implications of analysing fossil groups for which molecular data are unavailable.

[1] Milner Centre for Evolution, Department of Biology & Biochemistry, University of Bath, Bath, UK. [2] Vertebrates Division, Department of Life Sciences, Natural History Museum, Cromwell Road, London, UK. [3] School of Life Sciences, Joseph Banks Laboratories, College of Science, University of Lincoln, Lincoln, UK.
✉email: jwo22@bath.ac.uk; bssmaw@bath.ac.uk

Phylogenies are essential in many areas of biology[1], being widely utilised in evolutionary biology[2,3], ecology[4], conservation[5], parasitology[6] and medicine[7]. But what is the best way to produce an accurate phylogeny? Prior to the advent of molecular sequencing, morphology was the sole source of character data for phylogenetic inference in extant taxa[8]. Since the 1990s[9], however, the balance has shifted dramatically in favour of phylogenomic data[10].

Studies of homoplasy and convergence demonstrate that morphological similarity can sometimes be a poor guide to evolutionary relationships[11]. While some argue that molecules should invariably have primacy in phylogenetic inference[12], morphological and molecular data are often reciprocally illuminating, as shown in large-scale phylogenies of arthropods[13], reptiles and birds[14]. This balanced approach, acknowledging that both types of data have strengths, is now common in systematics[15,16]. While phylogenetic hypotheses derived from morphology are often supported by molecular data[17], molecules have also overturned many long-standing morphological hypotheses[18]. For example, phylogenomic analyses of placental mammals[19] have drastically altered the sequence of deep branching events traditionally supported by morphology[20]. Newly resulting mammal clades (e.g. Afrotheria, Atlantogenata, Boreoeutheria, Laurasiatheria)[21] are more congruent with their current geographic distributions, and have been named accordingly. Equally, molecular trees often conflict with each other, most notably when they are inferred using different sets of genes.

In the absence of known phylogenies, there can be no definitive assessment of the accuracy of branching patterns[22,23]. However, it is useful to evaluate conflicting trees using additional and independent criteria. Here we utilise two independent sources of data, namely biogeographic distributions and first stratigraphic occurrences. Before the cladistic revolution, biogeography was sometimes used to infer the relationships of extant taxa in combination with morphological data[24,25]. Although congruence with stratigraphy can be used as an ancillary criterion to choose between equally optimal trees for groups with a good fossil record, neither biogeographic[26] nor stratigraphic data[27–29] are routinely used to infer phylogeny today.

Since Wallace and Darwin, observations on the geographic distributions of species have underpinned the development of evolutionary theory[30]. Numerous studies have demonstrated non-random geographic patterns on evolutionary trees[31,32], and phylogenies are routinely used to test biogeographic hypotheses[33]. Here, we employ biogeographic congruence as an ancillary test of competing phylogenetic hypotheses using a sample of 48 matched pairs of morphological and molecular trees of animals and plants at multiple taxonomic levels. By using randomisation tests to compare the fit of the same biogeographic regions on paired morphological and molecular trees of the same taxa, our approach controls for differences in tree size and balance to the extent that these influence our indices of fit. We demonstrate that molecular phylogenies fit biogeographic data significantly better than their morphological counterparts. This difference in biogeographic congruence is not simply explained by differences in tree shape, tree resolution or when the trees were first published, although more recently published trees do tend to perform better. Ancillary tests using biogeographic congruence are shown to perform at least as well as existing tests based on stratigraphic congruence. We therefore propose that tests of biogeographic congruence, in combination with other tests, represent a useful way of evaluating competing evolutionary trees.

## Results

### Testing biogeographic congruence.
The process of summarising biogeographic data and assessing their fit onto trees is shown in Fig. 1 and described in detail in the Methods. Biogeographic occurrence data for extant taxa were compiled from the IUCN Red List of Threatened Species, Version 2019-2[34], the Global Biodiversity Information Facility (GBIF)[35] and The Reptile Database[36]. These distributions were used to define regions of shared taxa that summarised their present-day distributions, combining adjacent regions that contained identical taxon sets (see Supplementary Methods). Regional distributions were encoded in a matrix in the form of presence/absence scores for each taxon in each region. The fit of these biogeographic characters to both morphological and molecular trees was assessed using the ensemble consistency index (CI) and retention index (RI). However, our preferred index is a modified version of the homoplasy excess ratio[37], the biogeographic HER (bHER), derived from 10,000 random reassignments of biogeographic distribution data across terminals.

### Phylogenies tend to be significantly congruent with biogeography.
The overall congruence of phylogenies with biogeographic data was good: 54% of morphological and 65% of molecular trees had a significantly better fit than randomly permuted data at a $p$ value < 0.05 (and 69% of groups had one or both trees with a $p$ value < 0.05). Therefore, while biogeographic congruence for a minority of clades did not differ significantly from that expected by chance (e.g., Supplementary Fig. 1), most groups showed significant patterns that could be used to discriminate between trees. Biogeography and phylogeny are often thought to be correlated for major clades at large geographic scales (e.g., the distribution of placental mammal orders on continents[19]; Fig. 2a), and we find compelling evidence for similar patterns at other taxonomic levels and geographic scales (Fig. 2b, Supplementary Figs. 2, 3 and 4). Most biogeographic region matrices also had significantly non-random structure according to tree-independent permutation tail probability tests of pairwise character compatibility[38] (MCPTP tests: see Supplementary Methods). Our findings therefore support the use of biogeographic distribution data as an ancillary criterion for choosing between otherwise equally optimal trees, similar to the widespread practice adopted for stratigraphic congruence[39].

### Molecular trees are more congruent with biogeography than morphological trees.
Overall, biogeographic congruence was higher for our sample of molecular trees than for their morphological counterparts (Supplementary Fig. 5: means of 0.322 vs. 0.305, medians of 0.277 vs. 0.276 for CI; means of 0.263 vs. 0.228, medians of 0.211 vs. 0.183 for RI; Supplementary Fig. 6: means of 0.188 vs 0.121, medians of 0.153 vs. 0.108 for bHER). These differences were significant for all measures of biogeographic congruence according to Wilcoxon paired signed-rank tests (Table 1: CI; W = 685, Z = 2.22, rc = 0.384, $p$ value = 0.027, RI; W = 695, Z = 2.33, rc = 0.404, $p$ value = 0.0199, bHER; W = 888, Z = 3.08, rc = 0.51, $p$ value = 0.002) across the 48 pairs of trees, with molecular trees having greater congruence on average, according to each index (Fig. 3). Two-tailed sign tests also demonstrated that molecular trees had greater biogeographic congruence more often than their morphological counterparts (Fig. 4, Supplementary Table 1). Our samples of molecular and morphological trees did not differ significantly in their balance (how symmetrical or pectinate they were), the degree to which CI & RI differed from randomly permuted data or any stratigraphic congruence measure tested. The bHER is our preferred index, since it controls for tree size, balance and the number of biogeographic regions. Considering only groups with significantly structured (MCPTP test $p$ value < 0.05) region matrices (Supplementary Table 2), we recovered a similar result for bHER (W = 305, Z = 2.32, rc = 0.502, $p$ value = 0.019, $n$ = 28).

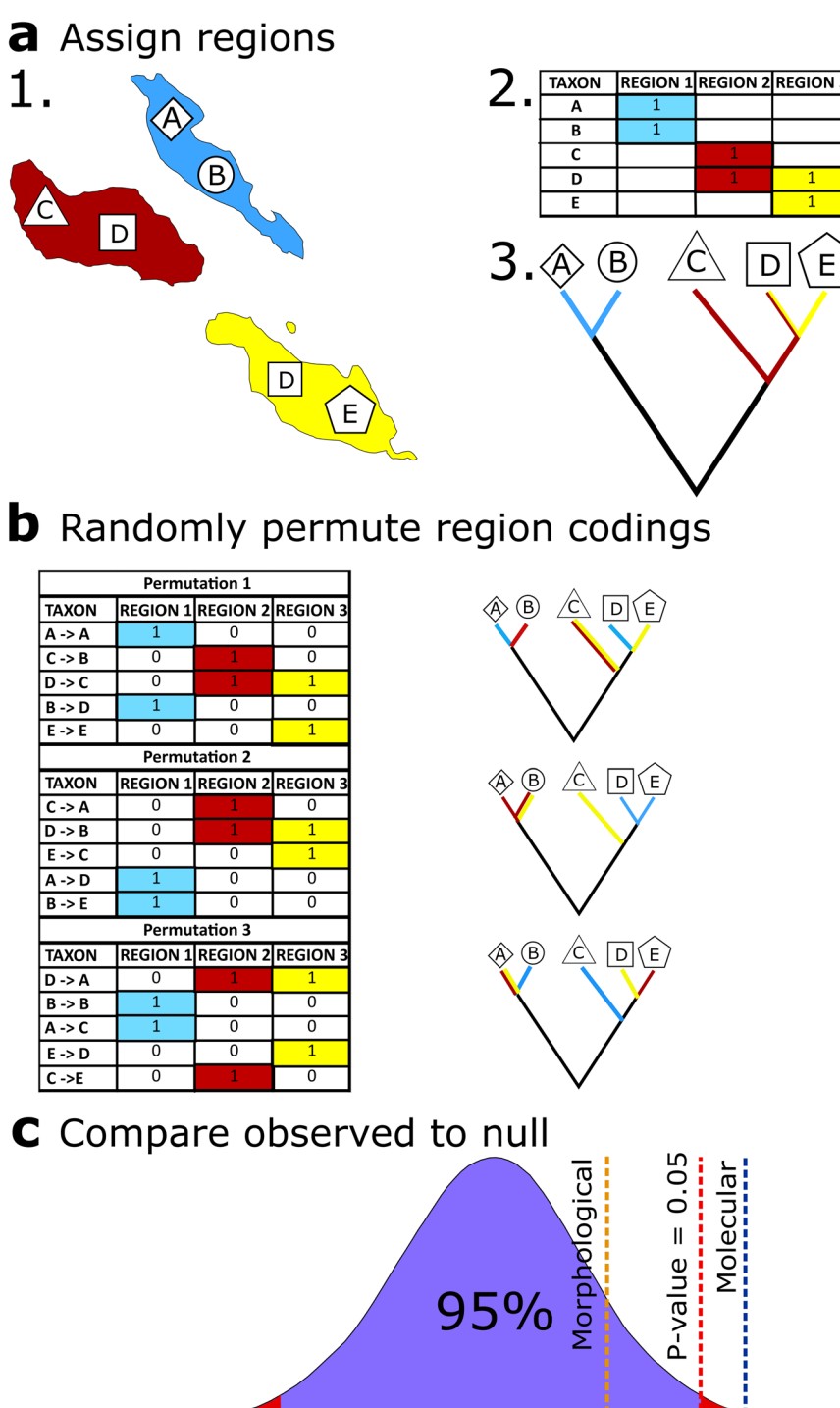

**Fig. 1 Testing the biogeographic congruence of phylogenetic trees. a** Defining biogeographic regions and coding taxon presences and absences. **1**. Occurrence data on the distribution of extant species is used to produce a list of biogeographic regions for the clade and to summarise ranges for the taxa in the published phylogenies. **2**. This distributional information is converted into a matrix of binary characters representing taxa in biogeographic regions, where 0 indicates the taxon is absent and 1 indicates the taxon is present. **3**. Characters in the occurrence matrix are mapped onto the morphological and molecular phylogeny selected for each clade, allowing standard measures of character fit (CI, RI) to be calculated for each tree. **b** Presence and absence codings in each matrix are randomly reassigned to taxa, keeping the presence and absence codings fixed for each row. Characters form the new randomly permuted matrix are mapped onto the original trees and both CI and RI are recalculated. The entire randomisation process is performed 10,000 times. **c** The 10,000 CI and RI values from matrices' biogeographic region reassignments form a null distribution of expected congruence values if taxa in the clade were randomly distributed in biogeographic regions. The observed CI and RI of region characters for a given tree is compared to the null distribution for that same tree to determine whether the observed biogeographic congruence value lies outside of the 95% confidence interval.

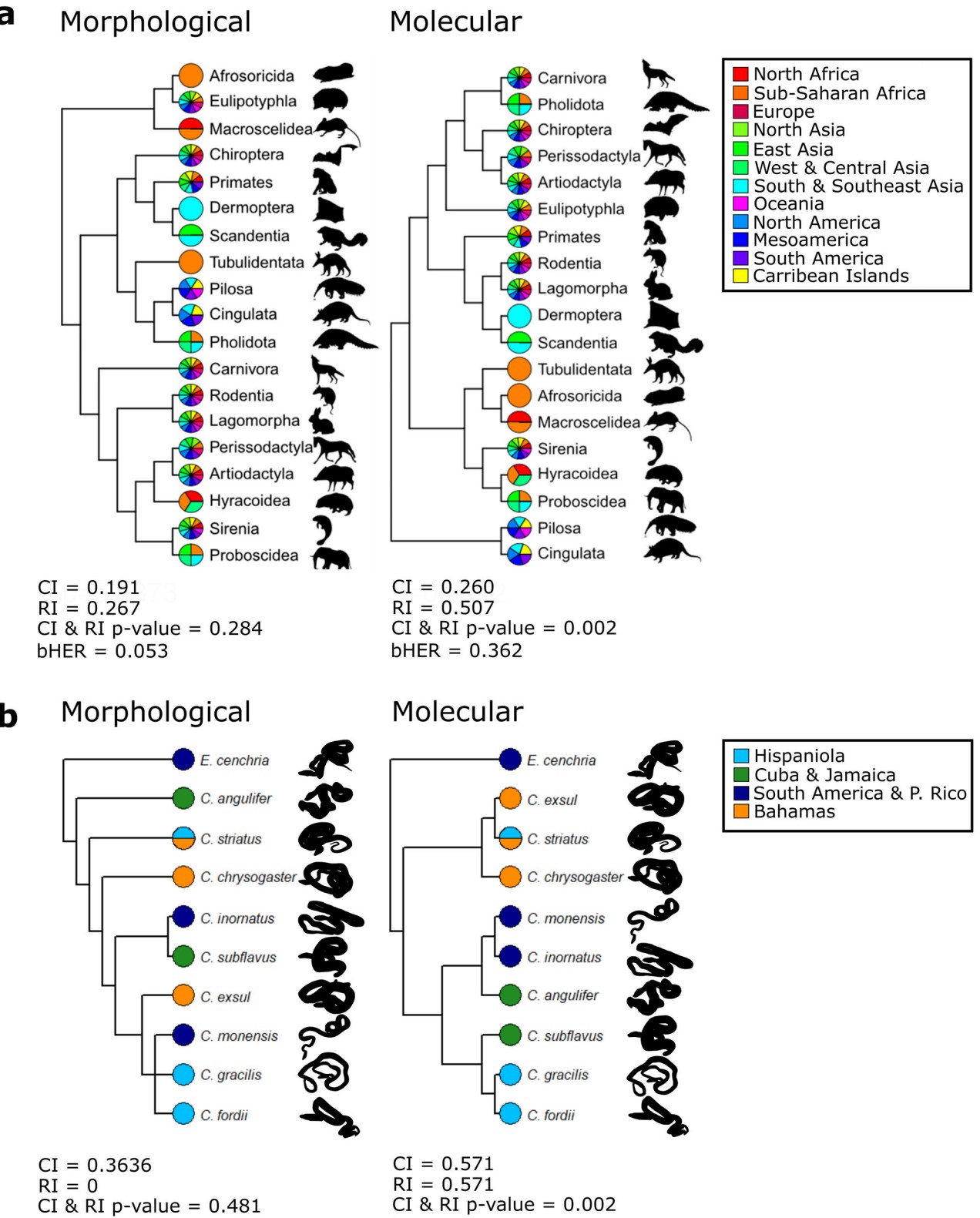

**Fig. 2 Biogeographic congruence in morphological and molecular phylogenies.** Binary biogeographic region characters mapped onto paired morphological and molecular phylogenies. **a** Placental mammals (Eutheria) from O'Leary et al. 2013[100]. **b** Caribbean boas (*Chilabothrus/Epicrates*), with the morphological tree taken from Kluge 1989[101] and the molecular tree taken from Tolson 1987[102]. Regions for which the terminal taxon is coded present are represented as coloured pie slices. Consistency index (CI), retention index (RI) and biogeographic HER (bHER) values given are for the matrix of biogeographic region presences and absences, while CI & RI *p* value is calculated using 10,000 randomly permuted region matrices.

**Table 1 Biogeographic and stratigraphic congruence of morphological and molecular phylogenies.**

| Metric | Median (morphological trees) | Median (molecular trees) | Wilcoxon signed-rank test statistic (W) | Z-score | Effect size (rc) | P value |
|---|---|---|---|---|---|---|
| Year | 2003 | 2003.5 | 59 | 0.947 | 0.297 | 0.362 |
| Size | 91 | 2221.5 | 1174 | 6.01 | 0.997 | $2.132 \times 10^{-14}$* |
| S/T | 4.852941 | 79.94878 | 1173 | 6 | 0.995 | $3.553 \times 10^{-14}$* |
| Res | 0.9168956 | 0.9928571 | 534 | 2.75 | 0.519 | 0.006035* |
| C | 0.2666665 | 0.2548875 | 508.5 | −0.815 | −0.135 | 0.4178 |
| CI | 0.276377 | 0.27705 | 685 | 2.22 | 0.384 | 0.027* |
| RI | 0.183279 | 0.2110125 | 695 | 2.33 | 0.404 | 0.0199* |
| CI & RI p value | 0.025897 | 0.013849 | 373 | −1.63 | −0.279 | 0.104 |
| bHER | 0.1078203 | 0.1533195 | 888 | 3.08 | 0.51 | 0.002* |
| SCI | 0.529 | 0.55 | 140.5 | 1.33 | 0.338 | 0.1913 |
| MSM* | 0.169 | 0.196 | 92 | −0.121 | −0.0316 | 0.9198 |
| GER | 0.571 | 0.588 | 91 | −0.523 | −0.133 | 0.6142 |
| GER* | 0.826 | 0.838 | 90 | 0.196 | 0.0526 | 0.8617 |

Results of paired Wilcoxon signed-rank tests on the two data partitions (morphological & molecular) for the following metrics: phylogeny publication year (Year), number of phylogenetic characters underpinning the source trees (Size), number of phylogenetic characters divided by number of taxa (S/T), proportion of resolved nodes (Res), Colless's index of tree balance (C), consistency index (CI), retention index (RI), probability of CI & RI values falling within the null distribution (CI & RI p value), biogeographic homoplasy excess ratio (bHER), stratigraphic consistency index (SCI), the modified Manhattan stratigraphic measure (MSM*), the gap excess ratio (GER) and the modified gap excess ratio (GER*). The sample size is 46 trees for SCI, MSM*, GER, GER* and 96 trees for all other metrics. Effect sizes were calculated using the matched-pairs rank biserial correlation coefficient. Statistically significant results at the 95% confidence interval are indicated with an asterisk. n = 23 biologically independent pairs of morphological and molecular phylogenies for all stratigraphic congruence metrics (SCI, MSM*, GER, GER*) and N = 48 biologically independent pairs of morphological and molecular phylogenies for all other metrics.

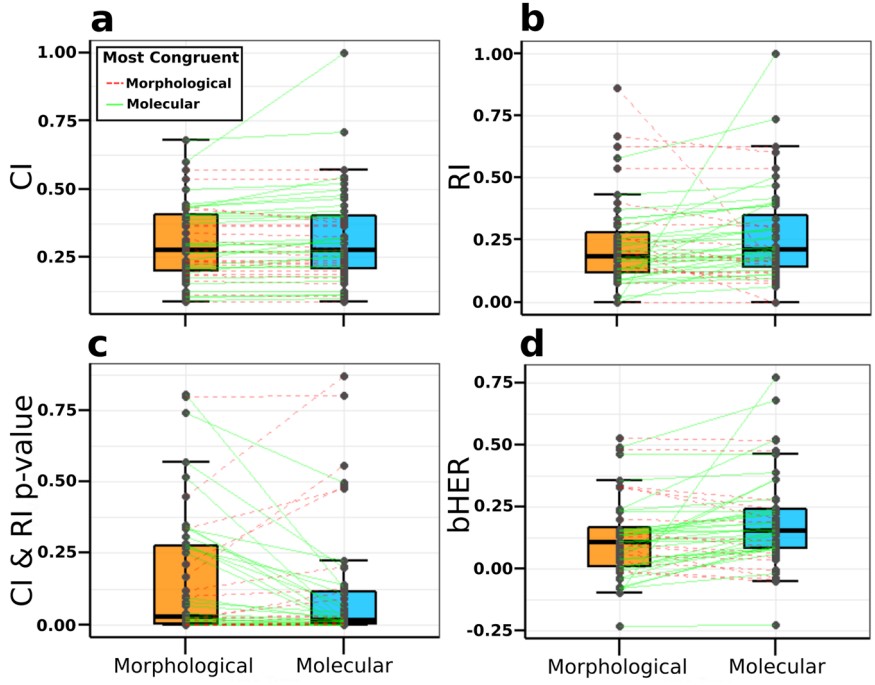

**Fig. 3 Differences in biogeographic congruence between morphological and molecular trees.** Boxplots of raw values and differences in values between morphological and molecular trees for the metrics of biogeographic congruence analysed in this study. **a** Consistency index (CI: W = 685, Z = 2.22, rc = 0.384, p value = 0.027). **b** Retention index (RI: W = 695, Z = 2.33, rc = 0.404, p value = 0.0199). **c** P values for the CI & RI random permutations (CI & RI p value: W = 373, Z = −1.63, rc = −0.279, p value =0.104). **d** Biogeographic HER (bHER: W = 888, Z = 3.08, rc = 0.51, p value = 0.002). Boxes delimit the upper and lower quartiles of the data, while central bars are median values. Whiskers delimit plus or minus 1.5 times the interquartile range, from the first and third quartiles. Coloured lines connected pairs of values from the same clade, where red dashed lines indicate the morphological tree is most biogeographically congruent and green solid lines indicate the molecular tree is most biogeographically congruent. N = 48 biologically independent pairs of morphological and molecular phylogenies.

In order to further ensure that the observed differences in congruence were not the result of conflating factors (Supplementary Table 3), we also modelled CI, RI and bHER as a function of tree type (morphological or molecular), clade root node age, tree balance (using Colless's index[40]), the number of geographic regions recognised, tree size (the number of terminal taxa), the ratio of characters to taxa (characters in the datasets used to generate the trees / the number of terminals), publication year and tree resolution expressed as the proportion of resolved nodes (number of internal nodes / (number of terminals – 2)). Multivariate linear regression models (Supplementary Table 4) supported publication year, number of biogeographic regions and

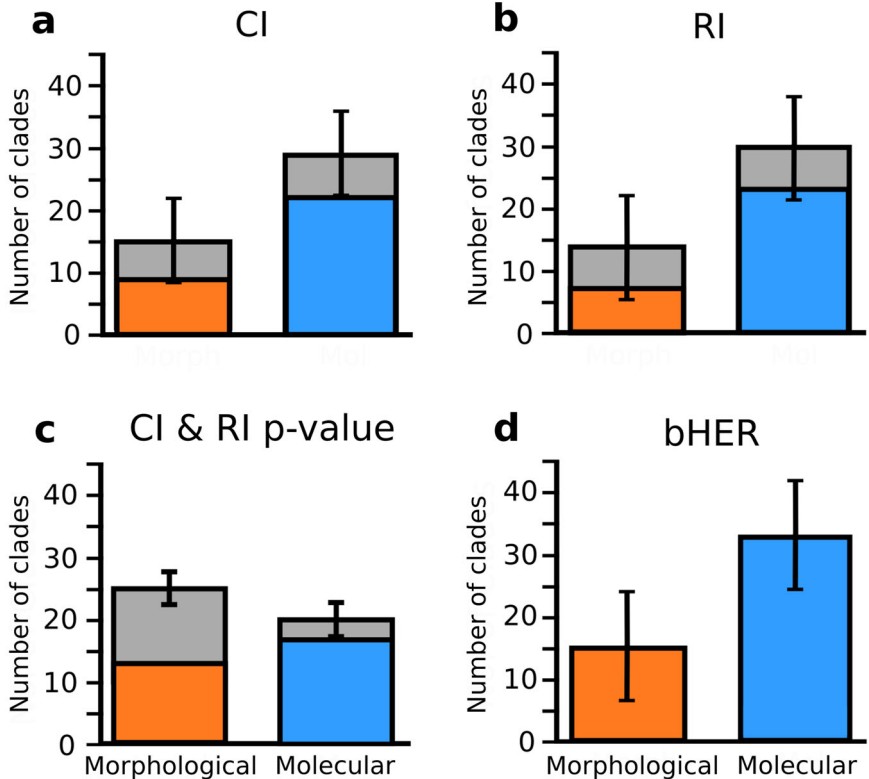

**Fig. 4 The number of morphological and molecular trees most congruent with biogeography.** Comparison of the number of trees in each sample (morphological or molecular) with a greater biogeographic fit than its counterpart. **a** Consistency index (CI), grey bars show totals for the whole sample, coloured bars indicate totals in the subset significantly different from the expected null (CI & RI p value < 0.05). **b** Retention index (RI), grey bars show totals for the whole sample, coloured bars indicate totals in the subset significantly different from the expected null (CI & RI p value <0.05). **c** P values for the CI & RI random permutations (CI & RI p value), where grey bars show totals for the whole sample, coloured bars are clades with values <0.05. **d** biogeographic HER (bHER), counts are for the whole dataset. Bars show the number of clades in each subset, with binomial confidence intervals calculated using the approach of Clopper and Pearson[103]. $N = 48$ biologically independent pairs of morphological and molecular phylogenies.

the proportion of resolved nodes together as the best predictors of bHER, while CI was best predicted by the combination of data type (whether the tree was morphological or molecular), the age of the root node, the number of biogeographic regions, the number of terminal taxa and the ratio of phylogenetic characters to taxa. In contrast, the number of region characters, along with the root node age and the proportion of resolved nodes were the best predictors of the RI. Despite this, residuals from weighted robust regression models and from minimum adequate models (MAMs) selected by the Akaike information criterion (AIC) showed a similar pattern to uncorrected values (Table 2), with CI and bHER demonstrating significantly greater biogeographic congruence for molecular trees (CI: W = 994, Z = 4.16, rc = 0.69, p value = $1.111 \times 10^{-5}$; bHER: W = 827, Z = 2.45, rc = 0.406, p value = 0.013). Morphological trees contained more polytomies (Supplementary Table 5) and significantly fewer resolved nodes (Table 1), but there was still a significant difference between molecular and morphological bHER when groups with polytomous morphological trees were omitted (n = 16, W = 179, Z = 2.12, rc = 0.603, p value = 0.01459).

Significant differences in bHER were also recovered comparing only groups with the same number of leaves in polytomies (n = 16, W = 115, Z = 2.43, rc = 0.691, p value = 0.01309), only groups where 75% or more of the nodes in both trees were resolved (n = 38, W = 537, Z = 2.41, rc = 0.449, p value = 0.01485) and groups which differed in their proportion of resolved nodes by 5% or less (n = 16, W = 144, Z = 1.97, rc = 0.516, p value = 0.04937). Additionally, CI values showed no evidence of any correlation with

the number of polytomies, number of branches in the polytomies or the proportion of resolved nodes (Supplementary Fig. 7). While bHER showed evidence of significant but weak negative correlations with the number of branches in polytomies (Supplementary Fig. 8b) and the proportion of resolved nodes (Supplementary Fig. 8c), molecular trees still showed significantly greater congruence when comparing residual bHER values in each case (number of branches in polytomies: W = 789, Z = 1.6, rc = 0.265, p value = 0.03895; proportion of resolved nodes: W = 838, Z = 2.56, rc = 0.425, p value = 0.009612).

Whilst taxonomic sampling and clade age are, by definition, the same for each pair of morphological and molecular trees in our compilation, clade age itself might be expected to influence biogeographic fit. Both RI and bHER were weakly positively correlated with the log of clade root node age (Supplementary Fig. 9: RI; $R^2 = 0.04437$, p value = 0.0394; bHER; $R^2 = 0.05894$, p value = 0.01716), indicating that phylogenies with earlier divergence times are more congruent with biogeography. In both cases residual values from linear regressions of fit metrics against log root node age still showed a significant difference between molecular and morphological trees (RI: W = 695, Z = 2.33, rc = 0.404, p value = 0.0199; bHER: W = 888, Z = 3.08, rc = 0.51, p value = 0.001684). In addition, differences in fit metrics between morphological and molecular trees showed no evidence of any correlation with log root node age (Supplementary Fig. 10). Any putative correlation between clade age and biogeographic fit is therefore insufficient to explain the differences between morphological and molecular trees observed here.

**Table 2 Biogeographic congruence metrics modelled by potential confounding variables.**

| Model | Linear Regression | | | Wilcoxon signed-rank test | | | |
|---|---|---|---|---|---|---|---|
| | AIC | $R^2$ | P value | W | Z-score | Effect size (rc) | P value |
| CI ~ Age + log(Regions) + log(Taxa) + S/T | −152.119 | 0.5543 | $4.1 \times 10^{-16}$ | 994 | 4.16 | 0.69 | $1.111 \times 10^{-5}$ |
| CI ~ Age + C + log(Regions) + log(Taxa) + S/T + Year + Res | −146.239 | 0.5397 | $5.683 \times 10^{-14}$ | 1040 | 4.64 | 0.769 | $5.818 \times 10^{-7}$ |
| bHER ~ log(Regions) + Year + Res | −63.981 | 0.1139 | 0.0027 | 827 | 2.45 | 0.406 | 0.01349 |
| bHER ~ Age + C + log(Regions) + log(Taxa) + S/T + Year + Res | −57.635 | 0.0894 | 0.0313 | 793 | 2.1 | 0.349 | 0.03511 |
| RI ~ Age + log(Regions) + Res | −55.5291 | 0.1336 | 0.0010 | 568 | 0.3 | 0.0509 | 0.768 |
| RI ~ Age + C + log(Regions) + log(Taxa) + S/T + Year + Res | −48.3497 | 0.1019 | 0.0199 | 529 | −0.605 | −0.1 | 0.5518 |

Results of models predicting the consistency index (CI), retention index (RI) and biogeographic homoplasy excess ratio (bHER) of geographic region characters from the age of the clade root (Age), Colless's index of tree balance (C), log of the number of regions (log(Regions)), log of the number of terminal taxa (log(Taxa)), number of phylogenetic characters divided by number of taxa (S/T), phylogeny publication year (Year) and the proportion of resolved nodes (Res). Both the model with all explanatory variables and the model with minimal Akaike information criterion (AIC) are given for each congruence measure. Wilcoxon signed-rank tests are between model residuals for morphological and molecular trees from weighted robust linear regression models and effect sizes were calculated using the matched-pairs rank biserial correlation coefficient (rc). N = 96 morphological and molecular phylogenies (regression models) and N = 48 biologically independent pairs of morphological and molecular phylogenies (Wilcoxon signed-rank tests).

**Morphological and molecular trees have similar stratigraphic congruence.** Of our 48 pairs of morphological and molecular trees, 23 had at least 50% of terminals with a fossil record, and these were assessed for stratigraphic congruence (Supplementary Table 6). Our preferred index is the modified gap excess ratio (GER*)[27], since it is relatively insensitive to differences in tree shape (balance), tree size, and the distribution of first occurrence dates (although the latter two variables are constant for each of our pairs). Morphological and molecular trees (Supplementary Fig. 11) had similar GER* values overall (0.774 and 0.780 respective means; 0.826 and 0.838 respective medians), and Wilcoxon signed-rank tests (Table 1) revealed no significant difference between the distributions of GER* values (W = 90, Z = 0.196, rc = 0.0526, p value = 0.8617). We note that the highest stratigraphic congruence occurred more frequently in morphological (n = 10) than molecular trees (n = 8) (Supplementary Fig. 12), but this difference was not significant (Supplementary Table 7: sign test; n = 23, p value = 0.21). We observed similar results for the gap excess ratio (Supplementary Fig. 13a: GER; W = 91, Z = −0.523, rc = −0.133, p value = 0.6142), stratigraphic consistency index (Supplementary Fig. 14a: SCI; W = 140.5, Z = 1.33, rc = 0.338, p value = 0.1913) and modified Manhattan stratigraphic measure (Supplementary Fig. 14b: MSM*; W = 92, Z = −0.121, rc = −0.0316, p value = 0.9198). Although the power of statistical tests was likely impacted by reduced sample size, tests of biogeographic congruence using Wilcoxon signed-rank tests (Supplementary Table 8) and sign tests (Supplementary Table 9) showed significant differences for bHER when carried out on only those clades included in the stratigraphic analyses.

**More recently published trees tend to be more biogeographically congruent.** The history of systematic research is characterised by greater volumes of data being analysed with increasingly sophisticated methods and models[41]. All other factors being equal, we might therefore expect phylogenetic accuracy to increase over research time[21]. Across all 96 morphological and molecular trees, we observed significant positive correlation between publication year and bHER ($r_s = 0.257$, p value = 0.012) and negative correlation between publication year and p values from our biogeographic CI and RI ($r_s = −0.284$, p value = 0.005). Hence, more recent trees tended to have higher biogeographic congruence (Supplementary Fig. 15, Supplementary Table 10). A similar pattern was found for the bHER of the morphological trees considered alone ($r_s = 0.292$, p value = 0.044), but was not

significant for the molecular trees alone (bHER; $r_s = 0.184$, p value = 0.210; CI & RI p values; $r_s = −0.274$, p value = 0.060). A significant minority (22 from 48) of our tree pairs had different publication dates, but we found no significant difference in the median publication years of the morphological and molecular partitions (Wilcoxon signed-rank W = 59, Z = 0.947, rc = 0.297, p value = 0.362). An overall improvement in phylogenetic accuracy with research time may be driven partially by analysing increasing volumes of data, both in terms of number of taxa and numbers of characters. However, this trend cannot explain adequately the observed differences in biogeographic fit between pairs of morphological and molecular trees, as publication year was found to be a poor predictor of biogeographic congruence metrics in most cases (Supplementary Table 4) and residuals from linear regressions of congruence metrics against publication year were still significantly higher for molecular trees in each case (Wilcoxon signed-rank test: CI; W = 769, Z = 2.5, rc = 0.423, p value = 0.01274, RI; W = 760, Z = 2.4, rc = 0.406, p value = 0.01673, bHER; W = 867, Z = 2.86, rc = 0.474, p value = 0.003649).

## Discussion

The observation that biogeographic congruence is significantly greater than expected by chance alone for most of our clades (69% had one or both trees with CI & RI p value < 0.005) supports the use of biogeographic data as an ancillary test of phylogenetic accuracy. Moreover, median biogeographic congruence for our 48 molecular trees was significantly higher than for their morphological counterparts and biogeographic congruence was not a function of tree size and balance. Indeed, if our results are representative, biogeographic distribution may be a better ancillary test than the established criterion of stratigraphic congruence. Stratigraphic congruence might also be contingent on the method used for tree inference. For example, morphological trees constructed using maximum parsimony often show greater stratigraphic congruence than their Bayesian equivalents[42], despite the increasing use of Bayesian methods with morphological data[43,44], although see[45,46]. In this study, our ability to distinguish between morphological and molecular trees was likely limited by a small sample size (n = 23).

Molecular data offer several advantages over morphology. Firstly, molecular characters can be acquired in vastly greater numbers and more readily than morphological ones, and often with less taxonomic expertise[47]. Secondly, published sequence data can be readily searched, repurposed and reanalysed alongside novel sequences. Despite efforts to systematically archive

morphological character matrices and character descriptions[48], there is as yet no way to automatically produce iteratively larger morphological matrices in a manner analogous to that possible for molecular data[49]. Both factors mean that it is often far easier to compile large molecular data sets than it is to compile equivalent volumes of morphological data. Thirdly, morphological systematists must make judgements concerning the homology of their characters and the way in which they are coded[50]. Morphological variation is unlikely to be atomised in precisely the same manner by different systematists[51], whereas it has been argued that a priori rules mitigate against subjectivity and promote repeatability in molecular systematics. Fourthly, a well-developed body of theory and empirical data facilitate sophisticated models of molecular evolution[52], while mathematical models for morphological evolution are still in their infancy[53,54].

Of course, molecular phylogenetics is not without its own problems, including issues of homology (orthology detection, alignment, saturation and homoplasy), the dangers of model misspecification and systematic bias. Moreover, paralogy, incomplete lineage sorting and horizontal gene transfer mean that even accurate gene trees may be incongruent with species trees. However, all other things being equal, where molecular and morphological data yield conflicting trees, our results suggest that molecular trees are likely to be more accurate. Phylogenetic signals across multiple gene alignments are typically much stronger, and lead to higher bootstrap branch support and posterior probabilities than signals from morphology[55]. Most morphological characters are binary and may be more prone to saturation than nucleotides and amino acids (assuming roughly equal rates of molecular and morphological character evolution). Many morphological characters are formulated to capture variation in different parts of the taxon sample. In so doing, however, they often incorporate assumptions about the way in which evolutionary transitions occurred. This is particularly true of characters whose states are logically contingent upon the states of others. For example, one character might code the presence or absence of a limb, while other characters might code for the morphology of bones within that limb. Where limbs are absent, these bone characters are often coded with "not applicable" scorings. Many morphological matrices therefore contain blocks of characters that are strongly conditionally dependent. However, morphological character matrices are, in theory, 'infinitely extensible' as newly discovered aspects of variation are accommodated in successive iterations by adding more characters and states. This approach to the accretion of morphological datasets might make characters less likely to show saturation through reversions to the same coded states but may make convergent gains more likely. This is particularly true if the initial hypotheses of transitions are incorrect. Convergence in morphological character states is common[56], even in characters that pass some of the conventional tests of homology[57] and have been hypothesised in the literature as homologous characters for decades[58].

While it is true that morphological trees tend to be less resolved, comparisons restricted to fully resolved trees have demonstrated that real incongruence in their primary phylogenetic signals[59] must account for the differing fits of morphological and molecular trees to biogeography. What we are unable to investigate further without access to the original data and comparative branch support metrics[60] is whether this incongruence is primarily due to lack of information or misleading information in morphological data. If, for example, incongruent relationships in morphological trees are less well supported by indices such as bootstrap[61] or Bremer support[62] than relationships which are congruent with biogeography, it would suggest that the biogeographic incongruence of morphological trees is partly attributable to a lack of strong signal in the morphological data.

Despite molecular trees typically showing greater biogeographic congruence, we found several cases where morphological trees have better fit than their molecular counterparts, such as dogs (Canidae), squirrels (Sciuridae), bats (Chiroptera), kangaroos (Macropodidae), conifers as a whole (Pinales) and pines (Pinaceae). However, in these cases, congruence values (and specifically bHER) only marginally favoured the morphological trees. Members of some these clades, such as conifers and bats, can disperse or travel over long distances and so may have large geographic ranges that limit the number of region characters and hence impact the power of our tests. Some morphological datasets may also contain characters that have evolved in response to particular environmental conditions (e.g., the pine dataset was based on cone morphology). This may increase congruence with biogeography when the regions within the clade's range broadly correspond with these environmental zones. Some clades (e.g., Canidae) were present in many more distinct biogeographic regions than the number of taxa in the dataset. As each region is defined by a unique grouping of taxa, a high number of regions relative to the number of taxa implies that the same taxa occur in different combinations in order to specify each distinct region. A 'mosaic pattern' of this type is likely to occur when at least some of the constituent taxa have fragmented rather than continuous distributions. This might, in turn, be indicative of frequent and rapid dispersal over long distances. Such patterns are common in many clades, particularly large mammals[63,64] which typically have wide-ranging distributions. Alternatively, or in addition, mosaic patterns might result from the rapid fragmentation of an original range. Since this occurs on much shallower timescales than the deeper divergences of the major branches in the phylogeny[65], the original biogeographic signal can be obscured.

Other problems that can impact accuracy, including long-branch attraction and incomplete lineage sorting, are not unique to morphological data. While simulations suggest that likelihood and Bayesian analyses are more resilient to some of these issues[66], such methods are increasingly being applied to morphological data. For some clades, particularly mammals, it might be possible to estimate the likelihood of biogeographic character saturation. However, this would require independent data on the rate of biogeographic transitions (from either direct observations or population genetics), along with time-calibrated phylogenies with scaled branch lengths. For most of the clades in this study such data do not exist and would require extensive effort to collect. More importantly, there is no reason why any such putative saturation effects should detrimentally impact biogeographic congruence for morphological trees more or less than their molecular counterparts. Therefore, while either morphological or molecular trees may show better congruence in a particular case, biogeographic congruence still provides a valuable ancillary test of phylogenetic accuracy.

The biogeographic distribution of extant species arises by two main processes: vicariance and dispersal[67]. Vicariance is the division of an ancestral area of sympatry by a physical barrier to create allopatric populations that may ultimately speciate, while dispersal is the migration or diffusion of individuals from some centre of endemism[68]. The relative importances of these two processes remain controversial and probably depend upon environment and time scale. Vicariance is often invoked as a result of the formation of land barriers such as mountains or oceans while dispersal is associated with repeated migrations away from a reservoir[69] or centre of endemism[70], as well as with biotic interchanges[71]. Species distribution patterns are unlikely to be purely vicariant or dispersive[72] and may be shaped by additional factors such as range expansions[73], migrations[74] and extinctions[75]. Regardless of which process dominated, we expect the geographic regions assessed here (which are analogous to the

areas that would form the basis of area cladograms[76]) to show some level of congruence with phylogeny and to yield nonrandom distributions. While we concede that all our indices would be likely to yield higher values for a purely vicariant than a purely dispersive pattern, there is no reason why morphological or molecular trees should be preferentially more congruent with either pattern. It is possible that selection pressures that cause similar adaptations to evolve in similar environments might result in a bias in favour of morphological trees where 'convergent' geographical transitions have occurred. However similar phenomena may also occur in molecular datasets. For example, there is increasing evidence that horizontal gene transfers have happened numerous times in green plants[77] and other eukaryotes[78]. Some of these genes are associated with traits that likely conferred a selective advantage in particular environments, such as vascular tissues in land plants, pathogen resistance and the C4 photosynthesis pathway in grasses, and herbivory in insects. Under certain circumstances, therefore, selection for traits expressed by horizontally transferred genes could also result in mitochondrial trees reflecting biogeography more closely than the true phylogeny. Determining the potential impact of these phenomena, as well as the roles of dispersal and vicariance in the specific biogeographic patterns seen here would require much more detailed analyses. It would necessitate combining independent population or observational data on biogeographic transitions with time-calibrated phylogenies at the species or population level. Such data and trees are lacking for most clades, and morphological phylogenies at this resolution are almost unheard of. While such work would be invaluable, it is vastly beyond the scope of this study and would prohibitively reduce our sample size of case studies.

Despite the superiority of molecular trees, the reciprocal illumination of morphological and molecular data and the simultaneous "total evidence" analysis of multiple data types remain instrumental in resolving the deep relationships of many otherwise recalcitrant clades including arthropods[17], echinoderms[79], angiosperms[80] and embryophytes[81]. Even the major revisions to the mammalian phylogeny supported by molecular analyses have prompted subsequent re-evaluation of morphological data. The latter have subsequently yielded results in broad agreement with phylogenomic trees. Biogeographic congruence of both morphological and molecular trees was found to improve over research time (publication date), indicating that the quality of morphological as well as molecular trees has improved. This is likely to have resulted not only from advances in methodology, but also a trend for increasing phylogenetic dataset size, regardless of the type of data being analysed. We also note the reciprocal illumination of published molecular and morphological phylogenies through research time, although the nature of this influence on subjective aspects of taxon choice, optimality criteria and character coding is difficult to assess. Molecular phylogenies often impact on new comparative morphological analyses (particularly by prompting the re-evaluation of hypotheses of homology) but morphological trees can also influence our understanding of molecular evolution and phylogeny. For example, several earlier multigene and genome-wide phylogenies of major arthropod groups yielded a clade comprising myriapods and chelicerates[82,83], a group so strikingly at odds with comparative morphological analyses that it was named "Paradoxapoda"[84]. Such findings prompted a re-evaluation of analytical models for sequence data as well as the adequacy of taxon sampling for deep and ancient divergences[85].

More generally, we believe that the continued importance of morphological data in phylogenetic analyses is assured. Not only is phylogenetics built on a legacy of morphological research but approximately 98% of species are extinct, and morphology remains the only source of data for exclusively fossil taxa[86].

Moreover, fossils often realise combinations of character states that are unknown from the extant biota[87], sample otherwise extinct or sparsely populated branches of the tree, and preserve the order in which character states have evolved, thereby enabling a better appreciation of evolutionary transitions (e.g., fish-tetrapod transition[88] or theropod-bird transition[89]). A better understanding of morphological evolution and fossilisation biases Sansom and Wills[90], as well as broader character sampling[91] will be key to obtaining more accurate molecular tree calibrations. Despite the development of increasingly sophisticated clock models[92], there is often a paucity of good fossil calibration dates[93]. We hope that our study will stimulate further ancillary biogeographic and stratigraphic tests of phylogenies inferred from a variety of morphological, molecular and combined data sets using different methodologies.

## Methods

**Dataset Compilation**. We initially obtained 106 animal and plant phylogenetic trees from 61 papers published between 1981 and 2015. These were reduced to 48 pairs of morphological and molecular trees for the same clades (Supplementary Table 11), derived from the same paper whenever possible. Phylogenies were taken from the main text of the paper where possible, with supplementary material only being used if trees were not present in the main paper. In cases where multiple morphological or molecular phylogenies were given, we used those preferred by the authors. If the authors expressed no preference, we selected trees which had the most taxa, most characters or were most resolved, in that order. Trees with the greatest possible overlap in taxon sets were selected, subsequently pruning unique leaves to yield identical taxon sets (46% of trees had different sources, 24% of trees had one or more taxa pruned, and these had a mean of 63% of leaves pruned). Most clades (73%) were terrestrial and freshwater vertebrates with strong patterns of endemism, but insect (13%) and plant (15%) clades were also included. Only 10% of clades contained any marine taxa, partly a function of the difficulties of accurately ascertaining and coding regions in these environments.

**Coding Biogeographic Distributions**. To assess biogeographic congruence, region characters summarising the distributions of taxa were defined from biogeographic occurrence data which could then be mapped onto phylogenies (Supplementary Fig. 16). Biogeographic data were obtained primarily from The IUCN Red List of Threatened Species, Version 2019-2[34] and checked using data from the Global Biodiversity Information Facility[35] where available. The Reptile Database[36] was used for the reptile clades in the study, which were frequently poorly represented in the IUCN and GBIF databases. Biogeographic data from these sources was then checked against any available data from the original publications. Biogeographic data were collected in two forms: taxon presences defined at the highest resolution of areas available (e.g., 'California', 'U.S.A.' or 'North America') and point occurrences. Point occurrences were synthesised into a list of presences for areas at the highest resolution of the online database. Our approach to coding was inclusive insofar as taxa known from multiple regions were recorded as present in all of these regions. For each clade, lists were combined to create a biogeographic character matrix of presence/absence characters for each recognised region (column). Taxa were scored "1" if present in and "0" if absent from the smallest discrete regions listed. If these regions were at different scales for different taxa, the larger region was broken up into its constituent subregions to match the finest scale represented, with taxa coded as present in the larger region also coded as present in all the constituent sub-regions. A matrix of characters, rather than a single multistate character, allowed for taxa that were observed from more than one region. Regions were then checked to ensure that none of them overlapped or were duplicates of the same geographic area. This yielded a full list of the least inclusive regions in which the members of the clade were found. As the areas being combined were often defined geopolitically or at the limited spatial resolution of our data, the regions derived from them were only biogeographically meaningful if they contained unique information about how taxa are grouped in space. Therefore, to avoid over-splitting of regions, we combined pairs of closest geographically neighbouring regions with identical taxon presence/absences into a single larger region and continued this process until all regions had unique taxon presence/absences. As it was not uncommon for biogeographic region matrices to contain more regions than taxa after this process (as a difference in presence for one taxon was sufficient to define a distinct region) we merged regions with single unique taxa (autapomorphic region characters) into their geographically closest neighbours.

To test whether the resulting biogeographic region matrices could potentially inform phylogenetic inferences, we assessed their non-random structure using matrix compatibility permutation tail probability (MCPTP) tests[38] (Supplementary Methods). Two characters are incompatible if it is not possible to map them onto the same evolutionary tree without homoplasy. The test statistic is therefore the number of compatibilities (viz incompatibilities) between all pairs of characters in a matrix. Applying this test to the biogeographic character matrices is a means of

assessing their congruent hierarchical signal (and thus the biogeographic information that they represent), in precisely the same manner as a parsimony PTP. Fewer incompatibilities indicate a more highly structured character matrix which is more likely to be phylogenetically informative. Significant nonrandom structure in the biogeographic data might be considered as a necessary prerequisite for using those same data as an ancillary test of the accuracy of trees inferred from different data types. If differences in biogeographic congruence are truly indicative of the relative accuracy of morphological and molecular trees, then such differences should also be evident when considering only those biogeographic matrices with significantly nonrandom (potentially phylogenetic) signal.

**Testing Biogeographic Congruence**. We assessed the fit of the biogeographic matrices onto both morphological and molecular trees using the ensemble consistency index (CI), ensemble retention index (RI) and biogeographic HER (bHER) (Supplementary Table 12). We note that the CI is biased by tree size, and by tree shape and balance with certain types of characters[94] (e.g., irreversible and ordered). We therefore also measured congruence using a modification of the homoplasy excess ratio (HER) of Archie[37]. Our biogeographic HER (bHER) was calculated by comparing the additional step length over and above the minimum necessary (the observed length for our data (L) minus the minimum possible given the number and nature of characters (MINL)) with the mean additional step length from lengths for biogeographically randomly permuted data (MEANNS) (randomly reassigning rows in the data matrix to the taxa 10,000 times, while holding tree topology constant). The bHER (or, more precisely, our modified MEANNS) therefore differed from the HER in its original form by permuting rows of the matrix across taxa (rather than the entries within each column separately) and by calculating the length of the original and permuted biogeographic matrices on the morphological or molecular tree (rather than inferring a tree from these data). By permuting rows of codes across taxa (rather than each column of data across taxa independently), we ensured that there were no unrealised or unlikely combinations of regional distribution patterns. Specifically, bHER = 1 - (L - MINL) / (MEANNS - MINL) (see Supplementary Methods for full details). A similar procedure was also used to produce a distribution of tree length values from randomly permuted biogeographic data, against which the original tree length could be compared to yield approximate $p$ values (the probability that a length as short or shorter could be observed for biogeographic data distributed at random on the tree). This is equivalent to a randomisation test for both CI and RI and will yield the same $p$ values for both metrics by definition. All analyses therefore accounted for the expected congruence if rows of region characters were randomly distributed across taxa. This was factored into how bHER was calculated, whilst for CI and RI it was controlled with an ancillary randomisation test. More specifically, this null expectation is factored into calculating MEANNS and therefore the scaling of the index. This ensured that, unlike CI and RI, bHER was already standardised relative to the expected fit of the region characters onto the tree of interest.

As most metrics were not normally distributed (Supplementary Table 13), nonparametric statistical tests were used in most cases. Correlations between biogeographic fit metrics and other variables of interest were assessed to determine whether confounding variables might affect our results. Breusch-Pagan tests indicated that the residuals from regressions between metrics of interest did not show significant heteroskedasticity in most but not all cases (Supplementary Table 14). Given that data might be non-normal, and relationships may be nonlinear, Spearman-rank correlation was preferred, with Pearson's correlations also being calculated on the data after the identification and removal of outliers. Five groups contained molecular datasets far larger than all others (more than 9000 characters) and were classed as outliers. Each metric was tested against the number of phylogenetic characters in the source dataset (size: Supplementary Fig. 17, Supplementary Table 15), the year in which the phylogeny was published (publication year: Supplementary Fig. 15, Supplementary Table 12), the number of terminal taxa (taxa: Supplementary Fig. 18, Supplementary Table 16), the ratio of region characters to terminal taxa (region characters/taxa: Supplementary Fig. 19, Supplementary Table 17) and the ratio of phylogenetic characters to terminal taxa (S/T: Supplementary Table 18). The bHER, CI, RI and the $p$ values from CI & RI randomisation tests for morphological and molecular tree samples were compared using two-tailed paired Wilcoxon signed-rank tests using 'wilcox.test' in R. In each case, the functions 'wilcoxonZ' and 'wilcoxonPairedRC' from the package 'rcompanion' were used to calculate Z-scores and effect sizes as given by the matched-pairs rank biserial correlation coefficient. In addition, two-tailed sign tests were used to test whether selecting the most biogeographically congruent tree in each pair resulted in significantly more molecular or morphological trees being chosen than expected by chance.

**Testing Stratigraphic Congruence**. Data on the fossil record of each of the 48 clades in this study were collated from the Fossilworks portal of the Palaeobiology database[95] (PBDB) and Benton 1993[96], as well as data within the source papers (Supplementary Methods). 23 Clades had published fossil data for at least 50% of their leaves, and so were judged suitable for tests of stratigraphic congruence. First and last occurrences for all taxa were assigned at the stage-level after O'Connor et al.[39], using the International stratigraphic chart[97], the Geologic Timescale 2004[98] and the GeoWhen database[99]. Low preservation potential and scarcity often ensure that first fossil occurrences lag behind true times of origin, while scarcity prior to the actual point of extinction mean that lineages are lost from the record

prematurely (the 'Signor-Lipps effect'). Where stratigraphy was unresolved at the stage level, taxa were therefore assigned to the first stage in the time interval given for their first occurrence and the last interval of the time period for their last occurrence. Stratigraphic congruence was assessed using several previously published and commonly utilised metrics, namely the stratigraphic consistency index (SCI), modified Manhattan stratigraphic measure (MSM*), the gap excess ratio and its modification (GER and GER*). The stratigraphic congruence of morphological and molecular trees was assessed using paired Wilcoxon signed-rank tests as well as sign tests, in a similar manner to that detailed for the biogeographic congruence tests.

**Reporting summary**. Further information on research design is available in the Nature Research Reporting Summary linked to this article.

## Data availability

The data that support the findings of this study are available on the website figshare (https://figshare.com/) with the identifier https://doi.org/10.6084/m9.figshare.c.5946358, in addition to being available from the authors upon request.

## Code availability

All custom scripts and programs used to calculate bHER, randomly permute region matrices and carry out MCPTP tests are available from the authors upon request.

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

## Acknowledgements
We thank Tim Astrop for useful discussions and suggestions related to plotting the data as well as Tamás Székely, Polly Russell and Catherine Klein for useful discussions. J.W.O., M.R. and M.A.W.'s work was funded by the John Templeton Foundation grants 61408 and 43915. M.A.W.'s work was funded by BBSRC grants BB/K015702/1 and BB/K006754/1, as well as BBSRC studentship 1923592.

## Author contributions
J.W.O. and M.A.W. conceived the study, devised tests of biogeographic congruence, developed the methods and theory, and wrote the paper. M.A.W. devised and scripted the bHER and other permutation tests. J.W.O. compiled the data, undertook all primary analyses and devised/drafted all figures. M.W. carried out the compatibility tests, analysed the data, and performed the simulations. M.W. and M.R. analysed data and contributed text to the introduction and discussion.

## Competing interests
The authors declare no competing interests.
