## [Peer Review File · Communications Biology]

Reviewers' comments:

Reviewer #1 (Remarks to the Author):

This manuscript examines the comparative power of morphological and molecular trees to resolve biogeographic relationships and their congruency with other kinds of data, including stratigraphic. They have assembled nearly 50 different paired morphological and molecular trees as points of comparison and have applied several kinds of analyses to them, including a variant of a widely used method for mapping trees onto biogeographic patterns.

I am convinced by nearly everything in the paper except for the author's main conclusion, that molecular trees fit biogeographic patterns better than morphological trees. The reason I am unconvinced is that the rate of biogeographic dispersal of most of the taxa in the data set (notably the mammals, which account for nearly half the tree pairs) is much, much higher than the temporal resolution of trees. Therefore, I am skeptical that the mapping of biogeography onto the trees represents much more than noise for either the morphological or molecular tree (indeed, I was surprised that so many trees passed the test having more congruence than expected by chance). I suspect that the reason that the molecular trees perform better (which they do by only a small degree) is because they are, on average, more completely resolved and therefore simply behave better with respect to any mapping, biogeographic or otherwise. The authors conclude much the same thing in their discussion (lines 178-181).

Let me explain why I am skeptical of the biogeographic component and at the same time suggest a strategy that would better convince me. Here I will speak almost exclusively about mammals, because I do not know the biogeographic histories of the other taxa well enough to have a good understanding of their dispersal rates. Mammals move long distances every chance they get. At any given interval of the Cenozoic, the number of intercontinental dispersals by mammals is usually very high with the exception of the most isolated continents like South America and Australia. Individual species and very closely related clades of species move back and forth across continents on time scales of thousands to hundreds of thousands of years. For example, the following species-level taxa have spread across Europe, Asia, and North America within the very short interval since those species originated: wolves (*Canis lupus*), red deer/wapiti (*Cervus elaphus*), red fox (*Vulpes vulpes*), arctic fox (*Alopex lagopus*), elk/moose (*Alces alces*), brown bear/grizzly (*Ursus arctos*), woolly mammoth (*Mammuthus primigenius*), and reindeer/caribou (*Rangifer tarandus*). A list not quite as long could be compiled of species-level dispersals between North and South America. In other words, the rate of intercontinental dispersals could be on the order of 1 per 250,000 branch length years. See for example, Van Der Made (2011) and Johnson et al. 2006 (citations below). To accurately map those dispersals onto a phylogenetic tree (in other words to get good congruence), one would need trees where each species is represented by two or three tips, and all species in a genus or family would need to be sampled. In contrast, the individual branch lengths of many of the trees are millions if not tens of millions of years long, which allows for the possibility that the biogeographic state has changed several times along the branch, impossible to sample from just one tip. This is apparent, for example, in the biogeographic coding of mammals in figure 2A where many of the tips radiated into several biogeographic areas. Note that these codings are an under-representation because they are based exclusively on modern distributions. Dermoptera, Pholidota, and Proboscidea, for example, should also be coded as North America because in the time represented by their branch on trees 2A they have inhabited that continent; similarly, Tubulidentata should be coded as occurring in North Africa, Europe and South & Southeast Asia. If the tree in 2A was resolved to full species level, one would potentially capture much more of the biogeographic pattern and the overall fit of both trees to biogeography would improve.

What I would expect is that biogeographic fit improves with taxonomic sampling of the tree. The closer one gets to a fully resolved species-level tree, the better the biogeographic fit will be (at least as far as mammals are concerned). One could easily test this supposition by comparing biogeographic fit to depth of common ancestry (i.e. age of the root node) or better yet median common ancestry (the median age of all the nodes on the tree). I suspect ones with a lot of recent nodes and short branch lengths (temporally speaking) will have better biogeographic fits than taxa with very deep divergences. I expect this to be true for both morphological and molecular trees,

with any difference between the two better explained by the number of polytomies than by data source. I also expect the congruence between the tree topologies to improve as median divergence gets smaller, although that is just a supposition. The data in the supplemental file may support this. I extracted the statistics for all the mammal trees and (hastily) divided them into those with deep divergences (Eutheria, Glires, Feliformia), medium divergences (Arctoidea, Phyllostomid 1, Phyllostomid 2, Megachiroptera, Chiroptera 1, & Chiroptera 2), and shallower divergences (all the other mammals). Average biogeographic CI for deep was 0.18, for medium was 0.21, and for shallow was 0.31. Note, however, that pattern for RI and HER did not support my supposition as clearly: RA 0.16/0.29/0.22 and HER 0.11/0.20/0.16.

Another relevant factor that will affect biogeographic congruence is the taxon-specific dispersal rate. Volant clades might disperse more readily than terrestrial ones, and large terrestrial animals might disperse more readily than small ones (my list above of species whose range spans several continents are all large mammals). Similarly, some clades may be better dispersers than others because of their ability to tolerate broader extremes (e.g., homeothermic mammals may disperse across high-latitude corridors more often than poikilothermic clades).

So for all of these reasons, I don't think most of trees in the data set have appropriate power to resolve biogeographic history and therefore I am skeptical of the assertion that molecular trees resolve it better. But I reiterate that I am convinced that the molecular trees on average will resolve nearly everything better, which is the main message that comes across in the discussion section of this paper, largely because they have a more fully resolved topology and, perhaps, because they have a more accurate topology. I'm inclined to the first interpretation because the actual difference in the summary biogeographic statistics for the two kinds of tree (CI, RI, HER) is very small, whereas I would expect it to be substantial if the topology of the molecular trees were substantially more accurate.

I find the paper quite interesting in its present form and I could see it being published even in light of my skepticism because the techniques applied by the authors are widely used at the same scale of analysis (with the caveat that I am usually equally skeptical about those papers) and because it is hard to imagine a better data set or a more careful analysis. However, I think the paper would be improved if some attempt was made to assess the expected rate of biogeographic dispersal in each clade and the ability of each tree to sample biogeographic transitions. An analysis of density of tip sampling and/or median branch length (or age of common ancestor) similar to those you did with time of publication, stratigraphic congruence, etc. would be a quite valuable addition because your message might become that molecular phylogenies with appropriately dense taxonomic sampling map to biogeography better (or conversely, you might prove that my assumptions are completely wrong).

Specific comments:

Lines 171-175: I generally agree with your characterization of the differences in coding morphological and molecular data, but note that discoveries based on full genome sequences show that drawing homologies between "genes" and aligning their sequences is much more ambiguous than it appeared to be 20 years ago when we could work only with sequences of single genes that had been extracted from a genome using methods that carried a lot of assumptions about homology. If you have ever tried to assemble and align a genome sequence, or even identify the locations of a known gene in a new genome of a taxon that was previously unsequenced you will know what I mean.

lines 181-183: I wonder about the statement that two-state morphological characters are more prone to saturation than nucleotides. If morphological characters were objectively constructed in the way that nucleotide characters are, then this statement would logically have to be true (depending on the rate of evolution in the morphological vs. molecular character of course). But morphological characters, especially binary ones, are usually constructed based on patterns of variation observed in a data set (as you rightly state in lines 169-173). Consequently, a morphological character is probably much less likely to revert states due to "double hits" as a molecular phylogeneticist would say. For example, imaging a scoring of tetrapod digits. Character 1 might have two states: 5 digits, loss of digit 5. Character 2 might have states: 4 digits, loss of

digit 4. Character 3 might have states: Digit 1 longer than digit 2, digit 1 shorter than digit 2. The morphological phylogeneticists will have chosen the characters and states based on the observed variation in digits among the taxa under consideration. It is more likely that evolution of digit number will progress from one character to the next (e.g., transition in character 1, then transition in character 2, then transition in character 3), leaving "not applicable" scorings behind in the earlier transitions rather than reverting to their ancestral state. Consequently, I think a morphological data set is much less prone to saturation, the reason being that its characters are conditionally dependent on one another and because the morphological character matrix is infinitely extendable as new taxa and new characters and states are added. A multivariate morphometric data set, on the other hand, behaves more like a molecular data set because the number of characters are fixed across the data set and every taxon has to have a value. These data sets can, I think, be more easily saturated (and certainly behave similarly to saturation in that divergence slows with time). That said, morphometric traits have an infinite number of states (presuming their measurement scale is composed of real numbers) and in theory (but not necessarily in biological practice) they can diverge infinitely. See discussion of this point in Polly, 2001.

Lines 195-199: I don't think the pattern you describe here results from an original range becoming fragmented over time, it results from frequent, rapid, and long distance dispersal of mammal taxa. Mammals move every chance they get. Even within the genus *Canis*, and within the very closely related clade making up wolves and coyotes, there have been many intercontinental dispersal events with the last tens to hundreds of thousands of years. Furthermore, many of these clades exhibit parallel dispersals between continents in very short time scales. For example, South American carnivorans contain particularly clear examples of parallel dispersals into the continent: two independent dispersals of weasels (*Neogale frenata*, *Neogale africana* + *felipei*), grison (*Galictis*), at least one dispersal of otters, at least one dispersal of skunks, at least three dispersals of canids (Coyote, grey fox, plus ancestor of South American canid radiation), and probably at least six dispersals of felids (cougar, jaguar, ocelot, jaguarundi, Geoffroy's cat, Margay). Thus I would say in the sentence 198-199 that "Taxa may have dispersed back and forth between continents many times, obscuring the biogeographic signal."

Supplementary Information file:

Lines 181-195: Higher in the supplemental file you showed that morphological trees have more polytomies on average than the molecular trees (and you discuss this observation in the main text). What effect does that have on consistency indices of biogeographic data? In the original analyses, I would assume that the polytomies arise from the application of consensus methods. If one mapped the original phylogenetic character data onto the consensus tree, the CI would drop compared to any one of the best fit trees. Do we expect the CI of an independent biogeographic data set to be worse if it is mapped onto a consensus tree with polytomies? I think the answer is yes, but I have not been able to completely convince myself of that. And is there a bias between the type of consensus method that gets applied to morphological versus molecular trees? For example, morphological phylogeneticists were historically prone to applying strict consensus whereas someone else might have been inclined to apply a majority rule method. Is it possible that a simple methodological bias like that might explain the difference in biogeographic resolving power of the two types of tree?

Line 195 (Table): note that there are two places in the table where the same number has been entered twice: *Andira*, Morphological, CI and *Iguanidae*, Morphological, RI

References mentioned:

Johnson, W.E., Eizirik, E., Pecon-Slattery, J., Murphy, W.J., Antunes, A., Teeling, E. and O'Brien, S.J., 2006. The late Miocene radiation of modern Felidae: a genetic assessment. *Science*, 311(5757), pp.73-77.

Polly, P. D. 2001. On morphological clocks and paleophylogeography: Towards a timescale for *Sorex* hybrid zones. *Genetica*, 112/113: 339-357.

Van der Made, J., 2011. Biogeography and climatic change as a context to human dispersal out of Africa and within Eurasia. *Quaternary Science Reviews*, 30(11-12), pp.1353-1367.

General disclaimer

By performing this review, I have accepted the possibility that it might be published along with the paper. Nevertheless, I disagree with the policy of publishing reviews on general principles: I have not had the time to polish my text for publication, nor is my work intended for an audience other than the authors and editor. By the time authors have revised the paper, my suggestions will be moot (either because the authors changed the paper or successfully rebutted my comments) and will therefore be unintelligible to anyone who did not see their original manuscript. Lastly, reviewers should not be given the freedom to insert their own unreviewed opinions and interpretations (like this one) into the scientific literature. Despite my disagreement with the policy, I feel that reviewing is an essential part of the scientific process and anyone who publishes papers reviewed by their peers should reciprocate by reviewing the papers of others, regardless of the journal's policy on publishing reviews.

Clade	CI	Morph RI	HER	CI	Mol RI
Cretaceous - Eocene					
Eutheria	0.19	0.27	0.05	0.26	0.51
Glires	0.24	0.12	0.15	0.24	0.12
Feliformia	0.11	0.09	0.14	0.11	0.11
	0.18	0.16	0.11	0.20	0.25
Eocene-Oligocene					
Arctoidea	0.23	0.17	-0.07	0.23	0.16
Phyllostomid Bats 1	0.09	0.33	0.14	0.10	0.39
Phyllostomid Bats 2	0.20	0.19	0.16	0.20	0.18
Megachiroptera	0.21	0.23	0.16	0.22	0.30
Chiroptera 1	0.32	0.67	0.46	0.28	0.60
Chiroptera 2	0.19	0.18	0.33	0.19	0.18
	0.21	0.29	0.20	0.20	0.30
Oligocene-Present					
Canidae	0.34	0.15	0.10	0.33	0.09
Ceboidea	0.28	0.26	0.09	0.25	0.14
Chyrsochloridae	0.30	0.08	0.01	0.30	0.10
Didelphidae	0.10	0.13	0.01	0.11	0.21
Didelphinae	0.13	0.14	0.11	0.13	0.15
Diprotodontia	0.43	0.58	0.49	0.55	0.74
Echymyidae	0.27	0.16	-0.04	0.27	0.17
Erinaceidae	0.39	0.02	0.14	0.40	0.06
Macropodidae	0.21	0.21	0.20	0.21	0.21
Mormoopidae	0.27	0.32	0.13	0.27	0.31
Plectonini	0.50	0.33	0.24	0.52	0.39
Sciuridae	0.43	0.40	0.33	0.39	0.30
Talpidae	0.40	0.12	0.24	0.40	0.13
	0.31	0.22	0.16	0.32	0.23

HER

0.36

0.15

0.19

0.24

0.09

0.22

0.19

0.22

0.46

0.20

0.23

0.07

-0.04

0.09

0.11

0.11

0.68

0.06

0.16

0.19

0.12

0.28

0.24

0.52

0.20

Reviewer #2 (Remarks to the Author):

Review of Oyston et al: Molecular phylogenies
For: Communications Biology

Fredrik Ronquist
2021-10-08

This manuscript analyzes whether molecular or morphological phylogenies are more accurate by comparing how well they match biogeography and stratigraphy. My opinion on the study is ambiguous. On one hand, I think the paper presents a fresh perspective on this question, and the elegant study design makes it attractive. On the other hand, I think most phylogeneticists regard this question as already settled. On balance, I am supportive of publication but I am not convinced that Communications Biology is the right outlet.

As mentioned in the paper, morphological characters are increasingly analyzed using stochastic models and either maximum likelihood or Bayesian inference. If one adopts this approach, then it is sufficient to just compare the branch support or posterior clade probabilities of the two trees to see which type of character is more informative about phylogeny. One can also analyze the morphological and molecular characters simultaneously in a combined analysis, and then investigate whether the total-evidence tree is more similar to the tree inferred using morphological characters only or molecular characters only. Without having made a comprehensive survey, I think there is a large number of published analyses of this type, most of them clearly showing that molecular characters are more informative about phylogeny.

What is rarely done is to use these types of parametric statistical approaches to ask whether morphological or molecular trees are more consistent with some external (non-morphological and non-molecular) evidence of phylogenetic relationships, such as stratigraphy or biogeography. An exception is perhaps the debate on "rocks and clocks" (see, for instance, Philosophical Transactions issue on this theme from 2016) but it is focused on dating implications rather than on topological accuracy, the phylogenetic relationships usually based largely on the molecular evidence in these studies.

In comparison to these types of analyses, the non-parametric methods used in the manuscript are more primitive but at the same time more versatile. The big advantage is that they allow one to investigate the questions asked in the paper without reanalyzing the original data. I think the permutation tests used in the paper are quite reasonable, even though you rarely see the CI, RI or HER indices used today. Nevertheless, I do suggest some slight improvements of the tests in the detailed comments below.

The biggest value of the paper is its clear demonstration of molecular trees being more congruent with biogeography than morphological trees. The increase in congruence over time, especially in morphological trees, is also noteworthy. Could this be because more recent morphological trees are influenced by molecular results and therefore more accurate? Given that morphological character coding is often criticized for being subjective, this seems like a real possibility. The paper does not comment on it but I think it would be interesting to see whether the data might be consistent with this idea.

One aspect that is not covered in the paper is the effect of the informativeness of the data. Is there any suggestion that clades with higher support are more congruent with biogeography than clades with lower support? The morphological trees contain more polytomies and have significantly fewer resolved nodes than the molecular trees (Tables S2, S16), and the authors show that the difference in biogeographic congruence remains after removing the groups with polytomous morphological trees (L116-119) and after controlling for the proportion of resolved nodes (Table 2). However, these results do suggest that biogeographic congruence may be correlated with branch support across both morphological and molecular trees. Is this the case? That is, are molecular trees more congruent with biogeography simply because the molecular data are more

informative about relationships? Or are morphological characters "misleading" about relationships, causing more conflicts with biogeography?

More detailed comments

L46. I think it is incorrect that morphological hypotheses are "sometimes" supported by molecular data (although you see many researchers expressing this idea). In fact, morphological hypotheses are usually supported by molecular data but we scientists tend to have a biased view of this because we get so excited about the cases when they do not (and they are not infrequent; I agree on that). I suggest replacing "sometimes" with "often", which I think is closer to the truth.

L58-59. "more commonly used to infer relationships" I think it should be made explicit that this is in comparison with stratigraphy. I do not think that biogeography has ever been more commonly used to infer relationship than morphology.

L62. Insert "today" at the end of this sentence?

L72. "consistent with the resolution of the occurrence data". I do not understand this at all. Please rephrase.

L147-150. This conclusion does not follow from what is stated in the preceding sentences. The conclusion should be linked to a multivariate analysis accounting for data volume and publication year but no such analysis is presented.

L158-162. This is not explained properly. The small sample size might explain why no significant effect was found. But the effect that "morphological trees tend to be more congruent with stratigraphy than molecular trees"? If this effect is true then we should see a significant effect in this study. I do not understand the logic here.

L171-177. I react to a few phrases here, in particular "empirical data on substitution, insertion and deletion rates inform expectations". Empirical data also inform expectations of how likely morphological characters are to evolve, so there is no clear qualitative distinction between morphological and molecular data here. And empirical data do not directly inform us about substitution, insertion and deletion rates; inferring those rates from empirical data is quite challenging, as it is to infer rates of morphological evolution. I think the paragraph should also mention data quantity; it is just a lot easier to generate large quantities of molecular data than it is to generate equivalent quantities of morphological data.

L179ff. Why is the effect of informativeness not examined in the paper? See general comment above.

L183-184. What is presumably meant is that the underlying hypotheses of homology have persisted in the literature and survived conventional tests. Please rephrase.

L178-205. This entire paragraph is very difficult to follow. Please consider rewriting it. See also next comment.

L195-197. I do not follow the reasoning here at all. Please rephrase.

L206-232. This section is quite speculative and wanders off topic, I think. I suggest a more compact text, focusing on the fact that regardless of whether biogeographic history is dominated by dispersal or vicariance, we expect to see some level of congruence between phylogeny and distributions under reasonable assumptions.

L229-232. It could be commented here that similar phenomena may occur also in molecular trees. For instance, it has been repeatedly observed that entire mitochondria are transferred across species boundaries where species are sympatric or parapatric, and that these transfers may be driven by selective advantages. Thus, there is definitely a potential for mitochondrial trees to reflect biogeography more closely than the true phylogeny.

L263-266. What data source was used for what group and why?

L269. The difference between "point occurrence" and "recorded sighting" is not clear; is this not the same thing?

L284-289. It is not clear why this test was used, and I do not recall seeing the results of these tests presented in the main paper.

L295-300. The symbols are not defined correctly here. L and MEANNS are the actual lengths and not the additional step lengths, as is obvious from the equation. Also "random data" here and elsewhere means "randomly permuted data"; please specify.

L306. "randomized tree length values" I think means "length values from randomly permuted biogeographic data".

L312ff. Permutation tests are only used for CI/RI. Why not for bHER? It should be straightforward to develop the same type of permutation test for bHER.

Figure 1: Why use colors both for taxa and for regions? It becomes really confusing; there must be a better way of demonstrating these relatively simple permutation tests. Maybe arrows showing how the rows of the matrix are reassigned to different taxa in each permutation? And not using colors for taxa?

Table 2: I do not understand the ordering of the models. Why not group the bHER and RI models instead of mixing rows with bHER and RI models?

Response to Reviewer Comments:

Molecular phylogenies map to biogeography better than morphological ones

Jack Oyston*(jwo22@bath.ac.uk), Mark Wilkinson, Marcello Ruta & Matthew A.

Wills*(bssmaw@bath.ac.uk)

We would firstly like to thank the reviewers for their time and for their detailed and insightful comments, which have prompted a number of additions and changes to the manuscript. We believe that these changes have improved our paper very substantially. We would also like to clarify and highlight a few aspects of the study that we believe may have been overlooked or misinterpreted by the reviewers. We very much hope that our overhauled paper makes these issues clearer.

Reviewer 1: Response to remarks to the author

The first reviewer states they are convinced by much of the work but skeptical of our main conclusion, that molecular trees fit biogeographic patterns better than molecular trees. This skepticism stems primarily from the fact many groups show high rates of dispersal relative to the temporal resolution of their trees. They cite examples from mammals which account for almost half (46%) of the pairs of trees in our dataset. The reviewer argues that many of the trees in our dataset simply have insufficient resolution to map most biogeographic patterns:

“To accurately map those dispersals onto a phylogenetic tree (in other words to get good congruence), one would need trees where each species is represented by two or three tips, and all species in a genus or family would need to be sampled. In contrast, the individual branch lengths of many of the trees are millions if not tens of millions of years long, which allows for the possibility that the biogeographic state has changed several times along the branch, impossible to sample from just one tip.”

We believe that our statistical results refute this claim. It is unclear (and in our opinion, arbitrary by definition) what level of taxonomic resolution is ‘good enough’, with the taxonomic resolution instead being dictated by the evolutionary questions being asked and the limitations of the data. However we believe the level of resolution furnished by our trees and regions matrices is sufficient to distinguish the broad biogeographic patterns necessary to address the questions we ask, namely, whether there are general differences in how any observed biogeographic patterns, at any scale, map onto published morphological and molecular trees. **The majority of our region matrices show statistically significant structure that justifies mapping onto trees.** If the taxonomic resolution of the trees was insufficient to test biogeographic patterns we would expect none of the trees to differ significantly from our expected null, derived by randomly assigning taxa to regions. Instead, the majority of groups show biogeographical mapping better than expected by chance onto at least one tree (69% of groups have

one or both trees with $p < 0.05$). There is also no reason that we can see why limited taxonomic resolution would bias molecular trees to show greater congruence with biogeography than morphological trees, as the resolution in terms of taxon composition is the same for the morphological and molecular trees being compared in each pair. For the reasons described above, we believe we can reasonably refute the suggestion that the patterns observed here could be produced by noise, and feel our more general, albeit less detailed and accurate approach is justified for the scale of the study.

Sampling multiple tips from each species and all species within each genus or higher taxon analysed is clearly extremely time intensive. For individual clades, it would require extensive additional sampling of taxa both in terms of sequencing and the collection and coding of morphological data. The collection of these data and inferring phylogenies from would be enormously time consuming work. These typically form the basis of entire programs of research in their own right, each focused on a particular taxonomic group. Implementing it on the scale of our study, with broad coverage over many taxonomic groups is an impossibility at the present time, and would likely require years of focused research. We think it highly unlikely that this is what the referee is suggesting. After all, our goal was to compare the biogeographical and stratigraphical congruence of morphological and molecular trees *as published*, and the spur for us to attempt this was no less than the phylogeny of mammals at a higher taxonomic level – precisely the group that the referee highlights.

We also note, in this context, that palaeontological trees have their own temporal as well as taxonomic sampling issues, but stratigraphic congruence is widely used to choose between otherwise equally optimal trees.

‘Note that these codings are an under-representation because they are based exclusively on modern distributions. Dermoptera, Pholidota, and Proboscidea, for example, should also be coded as North America because in the time represented by their branch on trees 2A they have inhabited that continent; similarly, Tubulidentia should be coded as occurring in North Africa, Europe and South & Southeast Asia.’

We acknowledge that our mapping of biogeographic patterns is limited by looking only at modern distributions and that the distributional patterns of taxa change over time. Ideally, with a well-preserved fossil record, one may be able to infer the past distributions of taxa and code these changes appropriately. This could either be by counting all regions that a taxon inhabits along its branch as the

author suggests, or accounting for distributional changes through time – looking at biogeographic congruence in time bins for example. However, we believe such analysis goes beyond the scope of the work set out in our study and is also difficult to implement for two main reasons. Firstly, we conducted analyses of stratigraphic congruence as part of our study. The incorporation of fossil data into estimates of biogeographic congruence would make both measures dependent on the quality and properties of the fossil record and therefore non-independent of each other. Firstly, the fossil record of many groups is often of insufficient quality to accurately determine biogeographic patterns independent of phylogeny (indeed, biogeographic patterns for extinct groups are often inferred based on available fossils assuming a particular phylogenetic framework). Most of the clades in the dataset were not, in our view, sufficiently represented by a fossil record to permit meaningful analyses of stratigraphic congruence (23 out of 48 clade pairs) (see O'Connor et al., 2016). Secondly, both the resolution of the record and the basis of inference for biogeographic patterns are different for data based on extant only occurrences versus those inferred from past distributions of fossils. The extent to which the two are comparable depends upon the group in question. For example, the pollen record shows a high degree of temporal resolution at higher taxonomic levels, but is of limited use in discerning distributional patterns at the levels of species or genera, as well as there being caveats with regards to pollen dispersal over long distances under certain conditions. Estimates of the past distributions of many vertebrate clades (particularly terrestrial vertebrates for which we have the best understanding of present-day distributions) is often highly dependent of the presence of taxa which may only be preserved in particular environments and times, most notably fossil Lagerstätten. As we acknowledge our region characters (indeed, any assessment of biogeographic distributions) are a generalised and imperfect representation of the actual true distributions. This being so, it is important that these representations are as standardised, replicable and comparable as possible. It is for these reasons that, although analysis of past distributional patterns would possibly be an interesting avenue for future research, we limited our analyses to present-day distributions of extant taxa only.

‘What I would expect is that biogeographic fit improves with taxonomic sampling of the tree. The closer one gets to a fully resolved species-level tree, the better the biogeographic fit will be (at least as far as mammals are concerned). One could easily test this supposition by comparing biogeographic fit to depth of common ancestry (i.e. age of the root node) or better yet median common ancestry (the median age of all the nodes on the tree). I suspect ones with a lot of

recent nodes and short branch lengths (temporally speaking) will have better biogeographic fits than taxa with very deep divergences.'

Trees were published between 1981 and 2016, and many of these were either not time calibrated or did not give branch lengths or node ages in the source publication. We do believe that this proposition is worth testing however. While analyses of both branch length and median node age would be promising avenues of future research (since these data would – like more inclusive phylogenies above – be prohibitively time-consuming to acquire) we present additional analyses on the correlation of fit metrics with depth of common ancestry (root node age). In all cases estimates of root node age were obtained from TimeTree (<http://www.timetree.org>).

As most trees were relatively young, with a modest number of much older trees, the log of the root node age was used to assess correlations. While CI and the p-values of CI/RI compared to null did not show any correlation with root node age, both RI and bHER were found to show a positive correlation with the log of clade root node age (Figure 1). This seems to suggest the opposite of Reviewer 1's supposition, namely, that phylogenies with earlier divergence times might show stronger congruence with biogeography, albeit this trend was weak (RI: $R^2 = 0.04437$, $p = 0.0394$; bHER: $R^2 = 0.05894$, $p = 0.01716$).

Figure 1: Scatterplots of log median age of each clades root node in millions of years (x) vs. biogeographical fit metrics (y). **A:** Consistency index values ($R^2 = 0.01974$, $p = 0.1722$), **B:** Retention index values ($R^2 = 0.04437$, $p = 0.0394$), **C:** p-values of CI & RI values falling within the null distribution for randomly permuted biogeographic data ($R^2 = 0.03642$, $p = 0.06252$), **D:** Biogeographical HER values ($R^2 = 0.05894$, $p = 0.01716$). Molecular trees in blue, morphological in orange. Significant linear regression slopes are shown as dashed lines.

Additionally, for all fit metrics, differences between morphological and molecular trees showed no evidence of any correlation with log root node age (Figure 2). Unsurprisingly given this, and the fact that each pair of morphological and molecular trees shares the same root node age (by definition), residual values from linear regressions of fit metrics against log root node age still showed a significant difference between molecular and morphological trees (RI: $V = 695$, $p = 0.0199$; bHER: $V = 888$, $p = 0.001684$). Therefore, while there appears to be some limited and weak evidence that older clades show greater biogeographic congruence (likely due to the fact that the resolution of the trees tends not to capture all distributional patterns at the species level), we find no support for differences in fit being explained by the level of taxonomic sampling, even if one accepts that this isn't controlled for by the taxon sets being identical in each comparison.

Figure 2: Scatterplots of log median age of each clades root node in millions of years (x) vs. differences in biogeographical fit metrics (molecular tree fit – morphological tree fit) (y). Datapoints for molecular trees as shown in blue, points for morphological trees are shown in orange. **A:** Consistency index values ($R^2 = 0.001122$, $p = 0.8212$), **B:** Retention index values ($R^2 = 0.02846$, $p = 0.2517$), **C:** p-values of CI & RI values falling within the null distribution for randomly permuted biogeographic data ($R^2 = 0.0006764$, $p = 0.8607$), **D:** Biogeographical HER values ($R^2 = 0.001277$, $p = 0.8094$).

'I expect this to be true for both morphological and molecular trees, with any difference between the two better explained by the number of polytomies than by data source.'

The first reviewer raises the possibility that the observed differences in biogeographical congruence between morphological and molecular trees might be accounted for by differences in tree resolution, as the number of polytomies ($V = 404.5$, $p\text{-value} = 0.02556$) and how many branches are part of polytomies ($V = 394.5$, $p\text{-value} = 0.0149$) were found to be significantly higher for the sample of molecular trees, while the fraction of nodes that are resolved in the tree ($V = 169$, $p\text{-value} = 0.006$) was found to be significantly lower.

Crucially, even if we remove all pairs of trees contain different numbers of branches in polytomies (a smaller sample size of 16 pairs of trees, all of which also showed 4 or fewer

polytomous branches in total), there is still significantly better stratigraphic congruence for molecular trees ($n = 16, V = 21, p = 0.01309$). The same result is also found when all pairs of trees with more than 25% unresolved nodes or ($n = 38, V = 46, p = 0.04937$) or groups which differed in their fraction of resolved nodes by more than 5% ($n = 17, V = 33, p = n = 16, V = 21, p = 0.01309$) were removed.

As the second reviewer correctly notes:

'The morphological trees contain more polytomies and have significantly fewer resolved nodes than the molecular trees (Tables S2, S16), and the authors show that the difference in biogeographic congruence remains after removing the groups with polytomous morphological trees (L116-119) and after controlling for the proportion of resolved nodes (Table 2).'

Additionally, significant differences were also recovered comparing only groups where 75% or more of the nodes in both trees were resolved ($n = 38, V = 46, p = 0.04937$) and groups which had a differed in their fraction of resolved nodes by 5% or less ($n = 17, V = 33, p = n = 16, V = 21, p = 0.01309$).

While the fraction of resolved nodes (Res) was retained in the preferred model predicting bHER from confounding variables (AIC = -63.981, $R^2 = 0.1139$, p-value = 0.0027) in the stepwise linear regression analysis, correlations were weak. Importantly, residuals from all models, including the bHER models which retained Res still showed a significant difference in favour of the molecular trees (Table 1)

Model	Linear Regression			Wilcoxon signed-rank test	
	AIC	R ²	p-value	test statistic (V)	p-value
CI ~ Age + log(Regions) + log(Taxa) + S/T	-152.1193	0.5543	4.1x10 ⁻¹⁶	3	3.553x10 ⁻¹⁴
CI ~ Age + Im + log(Regions) + log(Taxa) + S/T + Year + Res	-146.2395	0.5397	5.683x10 ⁻¹⁴	36	3.537x10 ⁻¹¹
bHER ~ log(Regions) + Year + Res	-63.981	0.1139	0.0027	800	0.0047
bHER ~ Age + Im + log(Regions) + log(Taxa) + S/T + Year + Res	-57.635	0.0894	0.0313	781	0.0477
RI ~ Age + log(Regions) + Res	-55.5291	0.1336	0.0010	617.5	0.0054
RI ~ Age + Im + log(Regions) + log(Taxa) + S/T + Year + Res	-48.3497	0.1019	0.0199	930	0.0003

Table 1: Results of models predicting measures of consistency index (CI), retention index (RI) and biogeographical homoplasy excess ratio (bHER) of geographic region characters from the age of the clade root (Age), Colless's index of tree balance (*Im*), log of the number of regions (log(Regions)), log of the number of terminal taxa (log(Taxa)), number of phylogenetic characters divided by number of taxa (S/T), phylogeny publication year (Year) and the proportion of resolved nodes (Res). Both the model with all explanatory variables and the model with minimal Akaike information criterion (AIC) are given for each congruence measure. Wilcoxon signed-rank tests are between model residuals for morphological and molecular trees in the sample.

Similar results are recovered in robust regression analyses of these combinations of variables as well as for residuals modelling bHER as a function of the fraction of resolved nodes alone (Supplemental Datasheet 1: $V = 534$, $p\text{-value} = 0.006035$). The number of polytomies themselves show no correlation with our preferred metric bHER (Figure 3A). There is a weak negative correlation between bHER and the number of taxa in polytomies (Figure 3B: grey dashed line $F = 5.0222$, $p\text{-value} = 0.02738$), as well as a positive correlation between bHER and the fraction of resolved nodes (Figure 3C; grey dashed line $F = 5.0917$, $p\text{-value} = 0.02636$). This appears to be a function of extreme heteroskedasticity in the data (3B: $BP = 6.835$, $p\text{-value} = 0.008939$, 3C: $BP = 4.5962$, $p\text{-value} = 0.03204$) as inverse variance weighting results in a much weaker observed slope (Figure 3B: black dashed line $F = 8.6809$, $p\text{-value} = 0.004054$, Figure 3C: black dashed line $F = 9.1603$, $p\text{-value} = 0.003191$). **Residuals from the robust regression model of bHER as a function of the number of characters in polytomies *still* show significantly greater congruence for molecular trees ($V = 789$, $p\text{-value} = 0.03895$).**

Figure 3: Scatterplots of bHER (y) vs different measures of tree resolution. Datapoints for molecular trees as shown in blue, points for morphological trees are shown in orange. **A:** Number of polytomies (x) vs. bHER (y) (robust linear regression $F = 2.8438$, $p\text{-value} = 0.09504$) **B:** Number of taxa in polytomies (x) vs. bHER (y) (robust linear regression $F = 5.0222$, $p\text{-value} = 0.02738$) **C:** Fraction of resolved nodes (x) vs. bHER (y) (robust linear regression $F = 5.0917$, $p\text{-value} = 0.02636$).

‘Another relevant factor that will affect biogeographic congruence is the taxon-specific dispersal rate. Volant clades might disperse more readily than terrestrial ones, and large terrestrial animals might disperse more readily than small ones (my list above of species whose range spans several continent are all large mammals). Similarly, some clades may be better

dispersers than others because of their ability to tolerate broader extremes (e.g., homeothermic mammals may disperse across high-latitude corridors more often than poikilothermic clades).'

We acknowledge that different taxa have different dispersal abilities, and that this can influence biogeographical congruence. **However, these differences are controlled by our design. In all cases, we have comparable morphological and molecular trees for the same leaf sets.** Whatever factors influence one tree (population structure, ecology, dispersal ability, origination date) equally (by definition) influence the other.

Nonetheless, we make reference to differential dispersal ability in the text.

'Members of some these clades, such as conifers and bats, are able to disperse or travel over long distances and so may have large geographic ranges that limit the number of region characters and hence impact the power of our tests. Some morphological datasets may also contain characters that have evolved in response to particular environmental conditions (e.g., the pine dataset was based on cone morphology).'

We also note, however, that 73% of clades in the dataset were terrestrial and freshwater vertebrates with relatively strong patterns of endemism. Only 10% contained one or more marine taxa (marine taxa might be expected to disperse more readily than terrestrial ones). While we included some volant clades (most notably bats), we did so only in cases where taxa within the clade also showed relatively restricted distributions and there was sufficient occurrence data available to reasonably infer ranges (which often were broad for the reasons the author describes). This appears justified, as many clades of bats pass Matrix Compatibility PTP tests ($p < 0.001$) and show relatively high bHER values compared to other clades, suggesting they show relatively strong biogeographic signal for both morphological *and* molecular trees. Although our dataset contains a wide range of taxa with different dispersal rates and environmental tolerances, we see no reason why this variability would bias the dataset as a whole towards showing greater congruence for molecular trees, or even why differences in these traits would be expected to have an effect on the relative fit of pairs of trees for the same taxa in the same clade.

'I reiterate that I am convinced that the molecular trees on average will resolve nearly everything better, which is the main message that comes across in the discussion section of this paper, largely because they have a more fully resolved topology and, perhaps, because they have a more accurate topology.'

We refer to the additional analyses described above, as well as our treatment of additional variables in the manuscript and supplementary information. We believe that these demonstrate why a difference in tree resolution is insufficient to account for the greater biogeographic congruence of molecular trees.

Not least in this regard, **analyses that utilise only fully resolved trees as published still show a difference.**

'I think the paper would be improved if some attempt was made to assess the expected rate of biogeographic dispersal in each clade and the ability of each tree to sample biogeographic transitions. An analysis of density of tip sampling and/or median branch length (or age of common ancestor) similar to those you did with time of publication, stratigraphic congruence, etc. would be a quite valuable addition.'

While analyses of tip sampling or branch lengths would require the collection of a significant amount of additional data, we have implemented analyses using the age of the common ancestor (see the earlier comment on root node ages).

It is impractical and beyond the scope of this work to assess the expected rate of biogeographic dispersal in 48 taxonomically, ecologically and morphologically diverse clades of organisms. Moreover data on dispersal ability is lacking for many clades and contentious for others. Similarly, sampling biogeographic transitions in more detail would almost certainly require some *a priori* knowledge of what those transitions are based on. Again, this necessitates information other than their modern distribution, which is also beyond the scope of this work. We would also suggest that it addresses a different question unrelated to the usefulness of biogeographic data as an additional and independent criterion to select between competing phylogenetic trees.

Reviewer 1 specific comments

Manuscript

Lines 171-175: 'I generally agree with your characterization of the differences in coding morphological and molecular data, but note that discoveries based on full genome sequences show that drawing homologies between "genes" and aligning their sequences is much more ambiguous than it appeared to be 20 years ago when we could work only with sequences of single genes that had been extracted from a genome using methods that carried a lot of assumptions about homology. If you have ever tried to assemble and align a genome sequence,

or even identify the locations of a known gene in a new genome of a taxon that was previously unsequenced you will know what I mean.'

We entirely agree, and confess that our original wording was ambiguous and has been clarified. We were referring specifically to the fact that nucleobases themselves are more readily identified (using automated methods) prior to alignment while the nature of morphological characters and even anatomical structures (particularly in unusual or extinct taxa) is often less clearly defined and has to be determined or intuited by the worker. **We have expanded on this statement to highlight some of these ambiguities in genome assembly and alignment as these difficulties are often overlooked and deserve to be emphasised.**

lines 181-183: I wonder about the statement that two-state morphological characters are more prone to saturation than nucleotides. If morphological characters were objectively constructed in the way that nucleotide characters are, then this statement would logically have to be true (depending on the rate of evolution in the morphological vs. molecular character of course). But morphological characters, especially binary ones, are usually constructed based on patterns of variation observed in a data set (as you rightly state in lines 169-173). Consequently, a morphological character is probably much less likely to revert states due to "double hits" as a molecular phylogeneticist would say.

We thank the reviewer on this point, and include a discussion of these issues in the revised manuscript, including his example of digit coding in tetrapod limbs. Our point was really about the potential for character saturation given the nature of character state spaces, and we clarify this issue in the text. As Reviewer 1 notes, if we assume that all types of characters evolve at similar rates, then two-state characters would be expected to reach saturation faster than multistate characters. We have clarified that this is assuming both are evolving at roughly equal rates, as it might be argued that faster rates of genetic evolution make nucleotides more likely to reach high saturation. The discussion on methods of nested character codings for morphological data that allow morphological character matrices to be extended and may protect against saturation had not occurred to us and is important. Whether this is actually the case or not depends upon the methods of coding employed by the researcher.

Lines 195-199: I don't think the pattern you describe here results from an original range becoming fragmented over time, it results from frequent, rapid, and long distance dispersal of

mammal taxa. Mammals move every chance they get. Even within the genus *Canis*, and within the very closely related clade making up wolves and coyotes, there have been many intercontinental dispersal events with the last tens to hundreds of thousands of years. Furthermore, many of these clades exhibit parallel dispersals between continents in very short time scales. For example, South American carnivorans contain particularly clear examples of parallel dispersals into the continent: two independent dispersals of weasels (*Neogale frenata*, *Neogale africana* + *felipei*), grison (*Galictis*), at least one dispersal of otters, at least one dispersal of skunks, at least three dispersals of canids (Coyote, grey fox, plus ancestor of South American canid radiation), and probably at least six dispersals of felids (cougar, jaguar, ocelot, jaguarundi, Geoffroy's cat, Margay). Thus I would say in the sentence 198-199 that "Taxa may have dispersed back and forth between continents many times, obscuring the biogeographic signal."

Both the fragmentation of ranges and the frequent, long-distance dispersal would act to obscure biogeographic signal. The reviewer makes a compelling argument for long distance dispersal being the primary cause, at least in these particular mammal clades. We take great pains to stress that the analyses we conducted are silent on the relative importance of vicariant patterns vs. dispersal. Identifying the likely cause of the patterns we discussed would go beyond the scope of the work and require more detailed study of specific clades, as well as detailed knowledge of the distribution, ecology and evolutionary history of those clades.

Supplementary Information file:

Lines 181-195: Higher in the supplemental file you showed that morphological trees have more polytomies on average than the molecular trees (and you discuss this observation in the main text). What effect does that have on consistency indices of biogeographic data? In the original analyses, I would assume that the polytomies arise from the application of consensus methods. If one mapped the original phylogenetic character data onto the consensus tree, the CI would drop compared to any one of the best fit trees. Do we expect the CI of an independent biogeographic data set to be worse if it is mapped onto a consensus tree with polytomies? I think the answer is yes, but I have not been able to completely convince myself of that.

Most compellingly, if all pairs of trees containing different numbers of polytomies (or branches in polytomies) are entirely omitted from our analyses, the significantly superior biogeographical congruence of molecular trees (as published) remains.

Moreover, as previously shown (Figure 3), while there is a suggestion of a very weak negative correlation between bHER and the number of polytomies, these correlations are not significant (3A, 3B). While the fraction of resolved nodes shows a significant weak correlation with bHER, residual values of bHER are still significantly greater for molecular trees (in other words, the difference in biogeographic congruence remains when the effect of polytomies on bHER is corrected for). CI shows similar patterns to bHER (Figure 4), that is, there is a suggestion of a slight decrease in CI with increasing number of polytomies but this effect is extremely weak, non-significant and the distributions of data show high heteroscedasticity.

Figure 4: scatterplots of CI (y) vs, different measures of tree resolution. Datapoints for molecular trees as shown in blue, points for morphological trees are shown in orange. **A:** number of polytomies (x) vs. CI (y) (robust linear regression $F = 3.8983$, $p\text{-value} = 0.05127$) **B:** log number of taxa in polytomies (x) vs. CI (y) (robust linear regression $F = 3.3472$, $p\text{-value} = 0.07049$) **C:** fraction of resolved nodes (x) vs. CI (y) (robust linear regression $F = 0.93274$, $p\text{-value} = 0.3366$).

And is there a bias between the type of consensus method that gets applied to morphological versus molecular trees? For example, morphological phylogeneticists were historically prone to applying strict consensus whereas someone else might have been inclined to apply a majority rule method. Is it possible that a simple methodological bias like that might explain the difference in biogeographic resolving power of the two types of tree?

We did not compile data on the type of consensus approach used, but we think that this is unlikely. Firstly, if strict vs. majority rule methods were responsible for the patterns we recovered we would expect differences in resolution or the number of polytomies to account for the differences in congruence between each independent pair of trees (and we have shown they do not: see above). We also think it is unlikely, although possible, that differences in the methodology used to construct the trees is responsible for the result, as the majority of both types were constructed under parsimony (45/48 morphological and 30/48 of the molecular). A further study to explore the impact of inference method on the biogeographic and stratigraphic congruence of both morphological and molecular trees would be well worth pursuing, but is beyond the scope of the present study. Here, we seek to test for differences in congruence for morphological and molecular trees *as published*.

Line 195 (Table): note that there are two places in the table where the same number has been entered twice: Andira, Morphological, CI and Iguanidae, Morphological, RI

This is a typo in the supplementary table rather than an error in the data itself, and has been corrected.

Reviewer 2: Response to remarks to the author

On one hand, I think the paper presents a fresh perspective on this question, and the elegant study design makes it attractive. On the other hand, I think most phylogeneticists regard this question as already settled.

We respectfully disagree with the second part of this statement, principally because no-one has actually tested what we set out to test in this paper. We reference tests of stratigraphic congruence as the closest equivalent in terms of an independent test of phylogeny using information that does not directly inform the construction of those same phylogenetic hypotheses. Moreover, a question can often be regarded as settled without good evidence. In this case, we are concerned less with the popularity, ease of use or applicability of morphological and molecular methods than we are with means to independently test competing hypotheses. As palaeontologists, our results are not especially welcome!

As mentioned in the paper, morphological characters are increasingly analyzed using stochastic models and either maximum likelihood or Bayesian inference. If one adopts this approach, then it is sufficient to just compare the branch support or posterior clade probabilities of the two trees to see which type of character is more informative about phylogeny. One can

also analyze the morphological and molecular characters simultaneously in a combined analysis, and then investigate whether the total-evidence tree is more similar to the tree inferred using morphological characters only or molecular characters only. Without having made a comprehensive survey, I think there is a large number of published analyses of this type, most of them clearly showing that molecular characters are more informative about phylogeny.

While this is an extremely valuable approach, it really provides a test of the internal consistency of signal within a data set, and not an independent test of the accuracy of a tree. Consistent subjective bias in the coding of morphological data could result in artificially inflated levels of internal consistency that have little bearing on the veracity of that signal *per se*. Again, our study aimed to assess the biogeographical and stratigraphic congruence of morphological and molecular trees as published, and the proposed design would address a different question.

We feel it is important to clarify that although there are numerous published studies comparing partitions of data or branch support, these data are not independent. Instead, it is assumed that when greater volumes of character data are in agreement with each other then phylogenetic inferences are likely to be more accurate. Comparing branch support is of limited use because branch support will almost certainly strongly correlate with the size of the dataset. Molecular trees typically have far higher branch support than morphological counterparts even when topologies are identical, simply because the number of characters underpinning them is orders of magnitude higher. We foresee problems in controlling for large differences in character numbers: more than a few hundred is unusual in morphological analyses, while less than a few thousand is uncommon in molecular studies. We could implement randomisation approaches to control for these differences, but this moves the focus to the nature of data rather than the nature of trees.

Analysing morphological and molecular characters together in a combined analysis and in separate partitions is useful when investigating the influence of morphological or molecular data on the total-evidence tree, but provides no insight into which partition (if any) is likely to be more accurate and indicative of the true pattern of relationships. Again, the influence of a subset of characters on a combined analysis of the whole dataset is likely to be primarily determined by the number of characters in that subset. In order to have any confidence one is not simply testing the effect of dataset size, one

must therefore either compare datasets of strictly equivalent size or use some independent measure of accuracy, as we have done here.

The increase in congruence over time, especially in morphological trees, is also noteworthy. Could this be because more recent morphological trees are influenced by molecular results and therefore more accurate? Given that morphological character coding is often criticized for being subjective, this seems like a real possibility. The paper does not comment on it but I think it would be interesting to see whether the data might be consistent with this idea.

This would be interesting to pursue, but does go quite substantially beyond the scope of the study and would also require us to know what the authors were thinking at the time, which is impossible. We believe that morphological and molecular phylogenetic analyses are entirely complementary and mutually benefit one another, with the influence of each one on the other being more iterative. In the manuscript we highlight several cases where morphological and molecular trees have helped to inform each other leading to new understandings of evolutionary relationships, as well as cases where conflicting hypotheses of relationships have been reconciled with the addition of new data and methodologies.

One aspect that is not covered in the paper is the effect of the informativeness of the data. Is there any suggestion that clades with higher support are more congruent with biogeography than clades with lower support? The morphological trees contain more polytomies and have significantly fewer resolved nodes than the molecular trees (Tables S2, S16), and the authors show that the difference in biogeographic congruence remains after removing the groups with polytomous morphological trees (L116-119) and after controlling for the proportion of resolved nodes (Table 2). However, these results do suggest that biogeographic congruence may be correlated with branch support across both morphological and molecular trees. Is this the case? That is, are molecular trees more congruent with biogeography simply because the molecular data are more informative about relationships? Or are morphological characters “misleading” about relationships, causing more conflicts with biogeography?

As discussed above, measures of support are likely to be strongly influenced by numbers of characters, and so almost certain to come out in favour of molecular trees simply by virtue of this fact. Producing TSI indices or other measures of support across all the trees in the study would necessitate

access to all of the underpinning matrices (not all of which are available) and is therefore impracticable. There are also issues with comparability. As the second reviewer underscores, measures of branch support can be readily compared if the methodology used to produce the trees is the same. In our case, pairs of morphological and molecular trees were sometimes inferred using different approaches, which makes comparing support non-trivial. There are a number of analyses that would be interesting to develop and implement that would attempt to circumvent this issue, but they are contingent on reanalyses of the original data that are beyond the scope of current work. They would also require very careful experimental design given that support measures vary across the tree.

More detailed comments

L46. I think it is incorrect that morphological hypotheses are “sometimes” supported by molecular data (although you see many researchers expressing this idea). In fact, morphological hypotheses are usually supported by molecular data but we scientists tend to have a biased view of this because we get so excited about the cases when they do not (and they are not infrequent; I agree on that). I suggest replacing “sometimes” with “often”, which I think is closer to the truth.

This is true, and we accept that our original wording overemphasised the conflict between morphological and molecular data (because these cases tend to be the ones of interest as well as providing the rationale for this work). We have changed the wording as per Reviewer 2's suggestions to avoid this pitfall.

L58-59. “more commonly used to infer relationships” I think it should be made explicit that this is in comparison with stratigraphy. I do not think that biogeography has ever been more commonly used to infer relationship than morphology.

Our original wording was unclear and has been changed. The reviewer is of course correct that biogeography has never been more commonly used to infer relationship than morphology. What was more common in the past was the use biogeography to infer relationships *in addition to* morphological data, particularly in an ancillary fashion or as part of an initial hypothesis that would inform morphological comparisons.

L62. Insert “today” at the end of this sentence?

Manuscript revised in accordance with the reviewer's recommendation

L72. “consistent with the resolution of the occurrence data”. I do not understand this at all. Please rephrase.

Changed to ‘These distributions were used to define regions of shared taxa which summarised their present-day distributional relationships, combining adjacent regions that contained identical taxon sets (see Methods and Supplementary Methods).’

L147-150. This conclusion does not follow from what is stated in the preceding sentences. The conclusion should be linked to a multivariate analysis accounting for data volume and publication year but no such analysis is presented.

It is quite true that we cannot state with confidence that the improvement in phylogenetic accuracy with research time is caused by increasing volumes of data. Rather, that is an observation based on our knowledge of the field. However, it *is* true that the correlations between fit metrics and numbers of characters underpinning the tree (size) cannot explain adequately the observed differences between pairs of morphological and molecular trees. This is because these differences persist when the effect of this correlation is accounted for. Similarly, the number of taxa cannot account for the effect as the pairs of trees being compared had identical leaf sets. We have therefore revised these lines to:

‘Hence, we find an overall improvement in phylogenetic accuracy with research time. This may be driven partially by analysing increasing volumes of data, both in terms of number of taxa and numbers of characters. However, this trend alone cannot explain adequately the observed differences between pairs of morphological and molecular trees.’

‘L158-162. This is not explained properly. The small sample size might explain why no significant effect was found. But the effect that “morphological trees tend to be more congruent with stratigraphy than molecular trees”? If this effect is true then we should see a significant effect in this study. I do not understand the logic here.’

This is an error, no significant difference in stratigraphic congruence was recovered (Manuscript Table 1), although a difference in biogeographic congruence was recovered for the same sample. Changed accordingly to correct this:

‘In this study, our ability to distinguish between morphological and molecular trees was likely limited by a small sample size ($n = 23$) which might explain why we did not detect a significant difference in stratigraphic congruence.’

L171-177. I react to a few phrases here, in particular “empirical data on substitution, insertion and deletion rates inform expectations”. Empirical data also inform expectations of how likely morphological characters are to evolve, so there is no clear qualitative distinction between morphological and molecular data here. And empirical data do not directly inform us about substitution, insertion and deletion rates; inferring those rates from empirical data is quite challenging, as it is to infer rates of morphological evolution. I think the paragraph should also mention data quantity; it is just a lot easier to generate large quantities of molecular data than it is to generate equivalent quantities of morphological data.

We generally agree with the reviewer’s point here, and admit that these phrases were overly simplistic and come across in the text as somewhat naive. We have therefore revised the text to include:

‘Of course molecular phylogenetics is not without potentially difficult problems, including issues of homology (orthology detection, alignment, saturation and homoplasy), the dangers of model misspecification and systematic bias, and that, as a result of paralogy, incomplete lineage sorting and horizontal gene transfer, even accurate gene trees may be incongruent with species trees. However, all other things being equal, where molecular and morphological data yield conflicting trees, our results suggest that molecular trees are likely to be more accurate.’

We agree fully with the reviewer’s point about generating large quantities of molecular data and have revised the text to include the following:

‘Firstly, molecular characters can be acquired in vastly greater numbers and more readily than morphological ones, and often with less taxonomic expertise. Secondly, published sequence data can be readily searched, repurposed and reanalysed alongside novel sequences. Despite efforts to systematically archive morphological character matrices and character descriptions, there is as yet no way to automatically produce iteratively larger morphological matrices in a manner analogous to that possible for molecular data. Both of these factors mean that it is often far easier to compile large molecular data sets than it is to compile equivalent volumes of morphological data.’

L179ff. Why is the effect of informativeness not examined in the paper? See general comment above.

By informativeness the reviewer is presumably referring to branch support, as we cannot think of another independent measure of ‘informativeness’ that this could be in reference to, besides (possibly) resolution. We therefore refer to our early response in justifying why tests of branch support were omitted.

L183-184. What is presumably meant is that the underlying hypotheses of homology have persisted in the literature and survived conventional tests. Please rephrase.

Revised to:

‘Convergence in morphological character states is common⁵², even in characters that pass some of the conventional tests of homology⁵³ and that have been hypothesised in the literature as homologous characters for decades⁵⁴.’

L178-205. This entire paragraph is very difficult to follow. Please consider rewriting it. See also next comment.

L195-197. I do not follow the reasoning here at all. Please rephrase.

This paragraph highlights a number of clades in our dataset for which the morphological tree shows greater biogeographic congruence. We therefore discuss factors which could lead to morphological data being more congruent, as well as factors that could obscure biogeographic signals and therefore introduce noise into the comparisons of biogeographic fit in certain cases. We have rewritten this paragraph to make these points more clearly:

‘Despite these expectations, we find several cases where morphological trees have better fit than their molecular counterparts, such as dogs (Canidae), squirrels (Sciuridae), bats (Chiroptera, Macropodidae), conifers as a whole (Pinales) and pines (Pinaceae). However, in these cases values (and specifically bHER) are similar for both trees in the pair and only slightly higher for morphology. Members of some of these clades, such as conifers and bats, are able to disperse or travel over long distances and so may have large geographic ranges which limit the number of region characters and hence impact the power of the tests employed here. Some morphological datasets may also contain many characters that have evolved in response to particular environmental conditions (e.g., the pine dataset was based on cone morphology), which may increase congruence with biogeography when the regions within the clade’s range broadly correspond to these environmental zones. Some clades (e.g., Canidae) had matrices with many more biogeographic region characters than the number of taxa in the dataset. As each region character defines a unique grouping of taxa, a high number of region characters relative to the number of taxa suggests a small number of taxa are found together in different combinations in different areas to give rise to a relatively high number of unique groupings of taxa. This ‘mosaic pattern’ of groupings suggests that at least some of the taxa in these clades have patchy rather than continuous distributions, which might indicate either frequent, rapid dispersal over long distances or the fragmentation of an original range over time shorter time scales than the divergence times of the taxa in the phylogenies used, obscuring the original biogeographic signal. Other problems that may impact accuracy, such as long branch attraction and incomplete lineage sorting, are not unique to morphological data. While simulations suggest that likelihood and Bayesian analyses are more resilient to some of these issues⁵⁵, these methods are increasingly being applied to morphological data. Therefore, while either morphological or molecular trees may show better fit in particular cases, biogeographical congruence still provides a valuable ancillary test of phylogenetic accuracy.’

L206-232. This section is quite speculative and wanders off topic, I think. I suggest a more compact text, focusing on the fact that regardless of whether biogeographic history is dominated by dispersal or vicariance, we expect to see some level of congruence between phylogeny and distributions under reasonable assumptions.

We agree this is the most relevant and important point to emphasise, but were mindful to explain a little of the background and arguments for why you would ‘expect to see some level of congruence between phylogeny and distributions under reasonable assumptions’, as this view is far from universally accepted. We have rewritten it as:

“The biogeographical distribution of extant species arises by two main processes: vicariance and dispersal⁵⁶. Vicariance is the division of an ancestral area of sympatry by a physical barrier to create allopatric populations that may ultimately speciate, while dispersal is the migration or diffusion of individuals from some centre of endemism⁵⁸. The relative importance of these two processes remains controversial and likely dependent on environment and time scale. Vicariance is often invoked in association with the formation of land barriers such as mountains⁵⁷, while dispersal is associated with repeated migrations away from a reservoir⁵⁹ or centre of endemism⁵⁸, as well as with biotic interchanges⁶⁰. Species distribution patterns are unlikely to be purely vicariant or dispersive⁶¹ and may be shaped by related phenomena such as range expansions⁶², migrations⁶³ and extinctions⁶⁴. Regardless of which process dominated biogeographic history, we expect the geographic regions assessed here (which are analogous to the areas that would form the basis of area cladograms⁶⁵) to show some level of congruence with phylogeny and to yield non-random distributions. While we concede that all of our indices would be likely to yield higher values for a purely vicariant than a purely dispersive pattern, there is no reason why morphological or molecular trees should be preferentially more congruent with either pattern.”

L229-232. It could be commented here that similar phenomena may occur also in molecular trees. For instance, it has been repeatedly observed that entire mitochondria are transferred across species boundaries where species are sympatric or parapatric, and that these transfers may be driven by selective advantages. Thus, there is definitely a potential for mitochondrial trees to reflect biogeography more closely than the true phylogeny.

As suggested, we have added the following to the revised manuscript:

“However similar phenomena may also occur in molecular datasets. Thus, there is increasing evidence that horizontal gene transfers have happened numerous times in green plants (Chen et al. 2021) and other eukaryotes (Schönknecht et al. 2014), some of which are associated with traits that likely conferred a selective advantage in particular environments, such as vascular tissues in land plants, pathogen resistance and the C4 photosynthesis pathway in grasses, and herbivory in insects. Selection for traits expressed by horizontally transferred genes could result in mitochondrial trees reflecting biogeography more closely than the true phylogeny in some cases.”

L263-266. What data source was used for what group and why?

We have added clarification detailing the sources used for each group. Sources were selected based on volume of data, resolution and taxonomic coverage for the group in question:

“Biogeographical data were obtained primarily from The IUCN Red List of Threatened Species, Version 2019-2 (<http://www.iucnredlist.org>) and checked using data from the Global Biodiversity Information Facility, 29th Dec 2019 (<https://www.gbif.org>) where available. The Reptile Database, 24 Dec 2019 (<http://www.reptile-database.org>) was used for the reptile clades in the study, which were frequently poorly represented in the IUCN and GBIF databases.”

L269. The difference between “point occurrence” and “recorded sighting” is not clear; is this not the same thing?

Mention of “recorded sighting” has been removed.

Recorded sightings could refer to sighting a taxon in a particular geographical location without giving latitude and longitude, but in practice all regions were defined based on point occurrences first and then validated against any coarser geographical data, so that they are the same in practice.

L284-289. It is not clear why this test was used, and I do not recall seeing the results of these tests presented in the main paper.

These results are mentioned in the results section headed 'Phylogenies tend to be significantly congruent with biogeography' lines 90-93:

'The majority (60%) of biogeographic region matrices had significantly non-random structure according to tree-independent, matrix compatibility permutation tail probability tests of character compatibility (MCPTP tests: Supplementary Table 14).'

This is a Matrix Compatibility PTP test. The test statistic is the number of compatibilities (viz incompatibilities) between all pairs of characters in a matrix. Applying this test to the biogeographic character matrices is a means of assessing congruent hierarchical signal in these matrices (and thus the biogeographic information they represent) [no different from a parsimony PTP]. We might reasonably expect that any difference between molecular and morphological trees in their congruence with biogeographic data (and the use of biogeography to test phylogeny inferred from these different data types) would make sense only if these data actually contain phylogenetic information and are not simply indistinguishable from random. Our rationale was to see if the pattern held up when we considered only those biogeographic matrices that had significantly non-random (potentially phylogenetic) signal. It is another way in which we explore the robustness of our results (akin to referee requests for some measure of the informativeness of the morphological and molecular data).

L295-300. The symbols are not defined correctly here. L and MEANNS are the actual lengths and not the additional step lengths, as is obvious from the equation. Also "random data" here and elsewhere means "randomly permuted data"; please specify.

L and MEANNS have now been defined correctly and all references to "random data" amended to specify "randomly permuted data".

L306. "randomized tree length values" I think means "length values from randomly permuted biogeographic data".

It does, and we have corrected accordingly

L312ff. Permutation tests are only used for CI/RI. Why not for bHER? It should be straightforward to develop the same type of permutation test for bHER.

The version of the HER implemented does employ the same type of permutation test (that's why we refer to it as bHER and not simply HER). It is simply that this is factored into calculating the MEANNS and therefore the scaling of the index. Moreover, rather than permuting values in each column separately and inferring a new topology (or topologies) with its own length (number of steps) for each

permutation, we permute rows of the data set across the *same* topology. Unlike CI and RI, a separate permutation test would be superfluous, since the bHER is already standardised relative to the expected fit of the region characters onto the tree of interest. CI and RI are not.

Figure 1: Why use colors both for taxa and for regions? It becomes really confusing; there must be a better way of demonstrating these relatively simple permutation tests. Maybe arrows showing how the rows of the matrix are reassigned to different taxa in each permutation? And not using colors for taxa?

As suggested, the figure has been amended, and now has arrows to show row reassignment and symbols for taxa

Table 2: I do not understand the ordering of the models. Why not group the bHER and RI models instead of mixing rows with bHER and RI models?

Models were ordered by AIC, but on reconsideration this is confusing and the reviewer's suggestion to group by metric and order by AIC within each metric group makes more sense. We have reordered the rows of the tables accordingly.

REVIEWERS' COMMENTS:

Reviewer #1 (Remarks to the Author):

The authors have done a thorough job addressing the concerns I raised in my review of the original paper. They have convincingly demonstrated that my hypothesis that morphological trees spuriously fit biogeographic data (or any data) better because they have more polytomies on average than molecular trees is wrong, a finding they explain thoughtfully in their rebuttal to reviewers and in their revised text. While I do still think that mammal biogeographic congruence would be better studied at finer taxonomic and temporal scales (and that one can objectively determine that level by drawing on independent data about rate of biogeographic transitions and estimating the likelihood of saturation of biogeographic characters relative to branch lengths on the tree), the authors have convinced me that the better congruence between biogeography molecular trees than morphological trees is genuinely due to a better statistical fit. I appreciate the care with which they explored the questions I raised. In my opinion, the paper is now ready to publish.

Reviewer #2 (Remarks to the Author):

The main comments of reviewer 1 are all valid concerns in historical biogeography. However, I find that the paired design used in this study provides a powerful way of addressing these concerns. It is true that the ability to correctly infer historical distribution patterns will be affected by the age of clades, the density of tip sampling, the dispersal propensity, and the degree to which the distributions of extant members are affected by recent extinction or human-assisted dispersal (among other things). However, all of these factors should affect the biogeographic fit of the molecular and morphological trees equally, since they cover the same groups. I think it is good that the authors added an analysis of the effect of clade age but I see no reason for additional changes to the manuscript to address these concerns.

A critical question is whether the results of the study are due to a lack of phylogenetic information or to misleading information in morphological trees (see also my previous review). The number of polytomies (that is, unresolved nodes) is a very rough measure of phylogenetic information content. Thus, I am not surprised that the number of resolved nodes does not seem to explain the results. I think it would be interesting to try to identify whether it is lack of information or misleading information that causes the relatively poorer performance of morphological trees. However, addressing this properly is not straightforward, and it would presumably require reanalysis of all the datasets (as pointed out by the authors in response to my previous comment about this), which would be an enormous task. I think it would be good at least if the paper acknowledges that this is an important question, and how it could be addressed.

Although I think it is true, as I stated in my review, that it has been convincingly shown that morphological data sets are on average less informative about phylogeny than molecular data sets, I think the authors have a point when they say that it is possible that the molecular data are pointing more strongly in a particular direction than the morphological data, but that this is the wrong direction. Thus, I accept that the approach explored by the authors is a valuable complement to the existing literature.

In response to my observation that the quality of morphological trees seems to increase over time, the rebuttal says "In the manuscript we highlight several cases where morphological and molecular trees have helped to inform each other leading to new understandings of evolutionary relationships, as well as cases where conflicting hypotheses of relationships have been reconciled with the addition of new data and methodologies." I do not remember seeing this in the original version of the paper. If it has been added, it helps address my question. But I still would like to see a direct reference in the text to the fact that the quality of morphological trees, as indicated by biogeographic congruence, seems to increase over time, and potential explanations for this.

L178-205 in original manuscript: I still have difficulties following the discussion where canids are

given as an example. Would it be possible to simplify further?

L284-289. Has this explanation been included in the text?

L312ff. Could a clarifying note about this be added to the paper?

Response to Final Referee Comments:

Molecular phylogenies map to biogeography better than morphological ones

Jack Oyston^{1*} (jwo22@bath.ac.uk), Mark Wilkinson², Marcello Ruta³ & Matthew A. Wills^{1*}
(bssmaw@bath.ac.uk)

¹ Milner Centre for Evolution, Department of Biology & Biochemistry, University of Bath, Bath, UK

² Vertebrates Division, Department of Life Sciences, Natural History Museum, Cromwell Road, London, UK

³ School of Life Sciences, Joseph Banks Laboratories, College of Science, University of Lincoln, Lincoln, UK

We thank the referees for taking the time to review our revised submission and response letter. Both reviewers were suitably convinced by our arguments and the additional analyses that we presented in response to their original suggestions. They raised some points for discussion and clarification, and we have addressed these in our final revised submission.

Reviewer 1: Revisions in response to final discussion points

“While I do still think that mammal biogeographic congruence would be better studied at finer taxonomic and temporal scales (and that one can objectively determine that level by drawing on independent data about rate of biogeographic transitions and estimating the likelihood of saturation of biogeographic characters relative to branch lengths on the tree), the authors have convinced me that the better congruence between biogeography molecular trees than morphological trees is genuinely due to a better statistical fit.”

We agree that there are advantages to this finer scale approach, although we maintain that such independent data is much easier to come by for well-studied groups of mammals and birds than it is for many other groups. The biogeographic congruence of mammals would certainly be better studied at finer taxonomic and temporal scales where possible, but this would necessarily be limited to only those clades where such data exist, or specific groups where the collection of new data for this purpose is the central aim. Our aim with this study was to assess whether general patterns exist in as broad a sample as we could. The fact that mammal clades dominate our sample simply reflects the fact that a lot of phylogenetic and biogeographic research has been conducted on mammals. As such, there could be significant advantages in future research that broadened the taxonomic range and resulted in a less mammal-dominated sample, even if it meant sticking to the straightforward, coarse-scale approach that we adopted. We see this as entirely complimentary and in no way a replacement for finer scale studies with a more restricted taxonomic focus. We have clarified our stance and acknowledged the referee’s point with the following addition to our discussion:

‘For some clades, particularly mammals, it might be possible to estimate the likelihood of biogeographical character saturation. However, this would require independent data on the rate of biogeographical transitions (from either direct observations or population genetics), along with time calibrated phylogenies with scaled branch lengths. For most of the clades in this study such data do not exist and would require extensive effort to collect. More importantly, there is

no reason why any such putative saturation effects should detrimentally impact biogeographical congruence for morphological trees more or less than their molecular counterparts.'

We also add the following when discussing the roles of vicariance and dispersal in producing biogeographic patterns:

'Determining the potential impact of these phenomena, as well as the roles of dispersal and vicariance in the specific biogeographic patterns seen here would require much more detailed analyses. It would necessitate combining independent population or observational data on biogeographic transitions with time-calibrated phylogenies at the species or population level. Such data and trees are lacking for most clades, and morphological phylogenies at this resolution are almost unheard of. While such work would be invaluable, it is vastly beyond the scope of this study and would prohibitively reduce our sample size of case studies.'

Reviewer 2: Revisions in response to final discussion points

"It is true that the ability to correctly infer historical distribution patterns will be affected by the age of clades, the density of tip sampling, the dispersal propensity, and the degree to which the distributions of extant members are affected by recent extinction or human-assisted dispersal (among other things). However, all of these factors should affect the biogeographic fit of the molecular and morphological trees equally, since they cover the same groups. I think it is good that the authors added an analysis of the effect of clade age but I see no reason for additional changes to the manuscript to address these concerns."

We have incorporated the analysis of the effect of clade age along with other potentially confounding variables into the results section and supplementary materials. As discussed with regards to referee one's point, we have acknowledged that all of these factors and more will affect the ability to infer historical distributional patterns and biogeographic congruence in an absolute sense. We also acknowledge that determining the extent and nature of these effects would require finer scale, detailed analyses of the type referee one prefers but agree with referee two that "all of these factors should affect the biogeographic fit of the molecular and morphological trees equally".

"A critical question is whether the results of the study are due to a lack of phylogenetic information or to misleading information in morphological trees (see also my previous review). The number of polytomies (that is, unresolved nodes) is a very rough measure of phylogenetic information content. Thus, I am not surprised that the number of resolved nodes does not seem to explain the results. I think it would be interesting to try to identify whether it is lack of information or misleading information that causes the relatively poorer performance of morphological trees. However, addressing this properly is not straightforward, and it would presumably require reanalysis of all the datasets (as pointed out by the authors in response to my previous comment about this), which would be an enormous task. I think it would be good at least if the paper acknowledges that this is an important question, and how it could be addressed.

We agree that this is indeed an important question, although as the referee points out such a reanalysis of all the data would be an enormous task and require careful thought about the branch support metrics being compared. We have added the following short paragraph to the discussion to acknowledge this question and suggest some comparisons of branch support that would address it.

'While it is true that morphological trees tend to be less resolved, comparisons restricted to fully resolved trees have demonstrated that real incongruence in their primary phylogenetic signals⁵⁹ must account for the differing fits of morphological and molecular trees to biogeography. What we are unable to investigate further without access to the original data and comparative branch support metrics⁶⁰ is whether this incongruence is primarily due to lack of information or misleading information in morphological data. If, for example, incongruent relationships in morphological trees are less well supported by indices such as bootstrap⁶¹ or Bremer support⁶² than relationships which are congruent with biogeography, it would suggest that the biogeographic incongruence of morphological trees is partly attributable to a lack of strong signal in the morphological data.'

"In response to my observation that the quality of morphological trees seems to increase over time, the rebuttal says "In the manuscript we highlight several cases where morphological and molecular trees have helped to inform each other leading to new understandings of evolutionary relationships, as well as cases where conflicting hypotheses of relationships have been reconciled with the addition of new data and methodologies." I do not remember seeing this in the original version of the paper. If it has

been added, it helps address my question. But I still would like to see a direct reference in the text to the fact that the quality of morphological trees, as indicated by biogeographic congruence, seems to increase over time, and potential explanations for this.”

We were referring to two sections of the manuscript in particular. The first is this section from the introduction:

‘While some argue that molecules should invariably have primacy in phylogenetic inference ¹², morphological and molecular data are often reciprocally illuminating, as shown in large-scale phylogenies of arthropods ¹³, reptiles and birds ¹⁴. This balanced approach, reflecting that neither source of data is problem free, is now common in systematics ^{15,16}. While phylogenetic hypotheses derived from morphology are often supported by molecular data ¹⁷, molecules have overturned many long-standing morphological hypotheses ¹⁸. For example, phylogenomic analyses of placental mammals ¹⁹ have drastically altered the sequence of deep branching events traditionally supported by morphology ²⁰. Newly resulting mammal clades (e.g. Afrotheria, Atlantogenata, Boreoeutheria, Laurasiatheria) ²¹ are more congruent with their current geographic distributions, and have been named accordingly. Equally, molecular trees often conflict with each other, most notably when they are inferred using different sets of genes.’

The second section, which is longer and more relevant to the referee’s point was in the discussion. We have therefore expanded and revised this section of the discussion to cite specific examples more explicitly, directly reference the improvement in the biogeographical congruence of morphological and molecular trees over time and offer some potential explanations for this general increase in fit.

‘Despite the superiority of molecular trees, the reciprocal illumination of morphological and molecular data and the simultaneous “total evidence” analysis of multiple data types remain instrumental in resolving the deep relationships of many otherwise recalcitrant clades including arthropods ¹⁷, echinoderms ⁷⁹, angiosperms ⁸⁰ and embryophytes ⁸¹. Even the major revisions to the mammalian phylogeny supported by molecular analyses have prompted subsequent re-evaluation of morphological data. The latter have subsequently yielded results in broad agreement with phylogenomic trees. Biogeographic congruence of both morphological and molecular trees was found to improve over research time (publication date), indicating that the quality of morphological as well as molecular trees has improved. This is likely to have resulted

not only from advances in methodology, but also a trend for increasing phylogenetic dataset size, regardless of the type of data being analysed. We also note the reciprocal illumination of published molecular and morphological phylogenies through research time, albeit the nature of this influence on subjective aspects of taxon choice, optimality criteria and character coding is difficult to assess. Molecular phylogenies often impact on new comparative morphological analyses (particularly by prompting the re-evaluation of hypotheses of homology) but morphological trees can also influence our understanding of molecular evolution and phylogeny. For example, several earlier multigene and genome-wide phylogenies of major arthropod groups yielded a clade comprising myriapods and chelicerates^{82,83} – a group so strikingly at odds with comparative morphological analyses that it was named “Paradoxapoda”⁸⁴. Such findings prompted a re-evaluation of analytical models for sequence data as well as the adequacy of taxon sampling for deep and ancient divergences⁸⁵.

“L178-205 in original manuscript: I still have difficulties following the discussion where canids are given as an example. Would it be possible to simplify further?”

The section is presenting potential explanations for why some clades differ from the general pattern (i.e., their morphological tree is more congruent with biogeography) and scenarios where biogeographical signal might be particularly weak or noisy in our analyses. We highlight how groups that contain taxa with large ranges will produce few region characters via our method and this will necessarily reduce the power of tests that assess the fit of those characters onto phylogenies. The second point is that there might be correlations between geography and environment that promote convergent evolution of the same traits, and this might introduce a ‘false’ congruence between morphological phylogenies and biogeography masking true phylogenetic signal. Lastly, the example of canids was given to illustrate that groups containing relatively few taxa with large ranges can have a very high number of biogeographic region characters (rather than few as in the first case we mention). This results from the same few taxa being found in different combinations, so the combinations define unique regions even though the same taxa are found in many different regions. This indicates a ‘mosaic’ pattern of fragmented distributions, because if the distributions were continuous, then the overlap in the

ranges of different taxa would reduce the number of distinct biogeographic regions. We have simplified and streamlined the section using canids as an example to hopefully get this point across better.

‘Despite these expectations, we found several cases where morphological trees have better fit than their molecular counterparts, such as dogs (Canidae), squirrels (Sciuridae), bats (Chiroptera), kangaroos (Macropodidae), conifers as a whole (Pinales) and pines (Pinaceae). However, in these cases, congruence values (and specifically bHER) only marginally favoured the morphological trees. Members of some these clades, such as conifers and bats, can disperse or travel over long distances and so may have large geographic ranges that limit the number of region characters and hence impact the power of our tests. Some morphological datasets may also contain characters that have evolved in response to particular environmental conditions (e.g., the pine dataset was based on cone morphology). This may increase congruence with biogeography when the regions within the clade’s range broadly correspond to these environmental zones. Some clades (e.g., Canidae) were present in many more distinct biogeographic regions than the number of taxa in the dataset. As each region is defined by a unique grouping of taxa, a high number of regions relative to the number of taxa implies that the same taxa occur in different combinations in order to specify each distinct region. A ‘mosaic pattern’ of this type is likely to occur when at least some of the constituent taxa have fragmented rather than continuous distributions. This might, in turn, be indicative of frequent and rapid dispersal over long distances.’

“L284-289. Has this explanation been included in the text?”

This is referring to the following text from our response letter:

L284-289. It is not clear why this test was used, and I do not recall seeing the results of these tests presented in the main paper.

These results are mentioned in the results section headed ‘Phylogenies tend to be significantly congruent with biogeography’ lines 90-93:

‘The majority (60%) of biogeographic region matrices had significantly non-random structure according to tree-independent, matrix compatibility permutation tail probability tests of character compatibility (MCPTP tests: Supplementary Table 14).’

This is Matrix Compatibility PTP - the test statistic is the number of compatibilities (viz incompatibilities) between all pairs of characters in a matrix. Applying this test to the biogeographic character matrices is

a means of assessing congruent hierarchical signal in these matrices (and thus the biogeographic information they represent), no different from a parsimony PTP. We might reasonably expect that any difference in the congruence of molecular and morphological trees with biogeographic data (and the use of biogeography to test phylogeny inferred from these different data types) would make sense only if these data actually contain phylogenetic information and are not simply indistinguishable from random. So, the rationale was to see if the pattern holds up when we consider only those biogeographic matrices that do indeed seem to have (potentially phylogenetic) signal. This is a further way in which we explore the robustness of our results. We have therefore included an explanation of the matrix compatibility PTP test in the methods section of the revised manuscript:

‘To test whether the resulting biogeographic region matrices could potentially inform phylogenetic inferences, we assessed their non-random structure using matrix compatibility permutation tail probability (MCPTP) tests ³⁸ (Supplementary Methods). Two characters are incompatible if it is not possible to map them onto the same evolutionary tree without homoplasy. The test statistic is therefore the number of compatibilities (viz incompatibilities) between all pairs of characters in a matrix. Applying this test to the biogeographic character matrices is a means of assessing their congruent hierarchical signal (and thus the biogeographic information that they represent), in precisely the same manner as a parsimony PTP. Fewer incompatibilities indicate a more highly structured character matrix which is more likely to be phylogenetically informative. Significant non-random structure in the biogeographic data might be considered as a necessary pre-requisite for using those same data as an ancillary test of the accuracy of trees inferred from different data types. If differences in biogeographic congruence are truly indicative of the relative accuracy of morphological and molecular trees, then such differences should also be evident when considering only those biogeographic matrices with significantly non-random (potentially phylogenetic) signal.’

“L312ff. Could a clarifying note about this be added to the paper?”

This is referring to the following text from our response letter:

L312ff. Permutation tests are only used for CI/RI. Why not for bHER? It should be straightforward to develop the same type of permutation test for bHER.

The version of the HER implemented does employ the same type of permutation test (that’s why we refer to it as bHER and not simply HER). However, given the way the HER is calculated this is factored

into the metric itself when calculating the MEANNS rather than permuting each column separately. Unlike CI and RI, a separate permutation test would be superfluous, since the bHER is already standardised relative to the expected fit of the region characters onto the tree of interest, while CI and RI are not.

We have clarified this distinction in the methods section with the following revised text:

'The bHER (or, more precisely, our modified MEANNS) therefore differed from the HER in its original form by permuting rows of the matrix across taxa (rather than the entries within each column separately) and by calculating the length of the original and permuted biogeographic matrices on the morphological or molecular tree (rather than inferring a tree from these data). By permuting rows of codes across taxa (rather than each column of data across taxa independently), we ensured that there were no unrealised or unlikely combinations of regional distribution patterns. Specifically, $bHER = 1 - (L - MINL) / (MEANNS - MINL)$ (see Supplementary Methods for full details). A similar procedure was also used to produce a distribution of tree length values from randomly permuted biogeographic data, against which the original tree length could be compared to yield approximate p-values (the probability that a length as short or shorter could be observed for biogeographic data distributed at random on the tree). This is equivalent to a randomisation test for both CI and RI and will yield the same p-values for both metrics by definition. All analyses therefore accounted for the expected congruence if rows of region characters were randomly distributed across taxa. This was factored into how bHER was calculated, whilst for CI and RI it was controlled with an ancillary randomisation test. More specifically, this null expectation is factored into calculating MEANNS and therefore the scaling of the index. This ensured that, unlike CI and RI, bHER was already standardised relative to the expected fit of the region characters onto the tree of interest.'